# Microtubule rescue at midzone edges promotes overlap stability and prevents spindle collapse during anaphase B

Manuel Lera-Ramirez[1]*, François J Nédélec[2]*, Phong T Tran[1,3]*

[1]Institut Curie, PSL Research University, Sorbonne Université, CNRS UMR144, Paris, France; [2]Sainsbury Laboratory, Cambridge University, Cambridge, United Kingdom; [3]University of Pennsylvania, Department of Cell and Developmental Biology, Philadelphia, United States

**Abstract** During anaphase B, molecular motors slide interpolar microtubules to elongate the mitotic spindle, contributing to the separation of chromosomes. However, sliding of antiparallel microtubules reduces their overlap, which may lead to spindle breakage, unless microtubules grow to compensate sliding. How sliding and growth are coordinated is still poorly understood. In this study, we have used the fission yeast *S. pombe* to measure microtubule dynamics during anaphase B. We report that the coordination of microtubule growth and sliding relies on promoting rescues at the midzone edges. This makes microtubules stable from pole to midzone, while their distal parts including the plus ends alternate between assembly and disassembly. Consequently, the midzone keeps a constant length throughout anaphase, enabling sustained sliding without the need for a precise regulation of microtubule growth speed. Additionally, we found that in *S. pombe*, which undergoes closed mitosis, microtubule growth speed decreases when the nuclear membrane wraps around the spindle midzone.

**\*For correspondence:**
manuel.lera-ramirez@curie.fr (ML-R);
francois.nedelec@slcu.cam.ac.uk (FJN);
phong.tran@curie.fr (PTT)

**Competing interest:** The authors declare that no competing interests exist.

## Editor's evaluation

This study carefully quantifies microtubule dynamics during anaphase in the fission yeast *S. pombe*. The high quality data revealed two novel observations: that microtubule rescue occurs preferentially at the edge of the midzone and that microtubule growth speed decreases when the nuclear membrane wraps around the spindle midzone in late anaphase. This sheds additional light on the interplay between the nuclear membrane and the midspindle in closed mitosis, and this study will be of interest to cell biologists studying spindle dynamics and mitosis.

## Introduction

The mitotic spindle is a bipolar assembly of microtubules, motors, and microtubule-associated proteins (MAPs), that orchestrates chromosome segregation. During prophase and metaphase, kinetochore microtubules capture and biorient chromosomes, and in anaphase A they transport chromosomes from the cell equator to the spindle poles. During anaphase B, the spindle elongates to further separate the chromatids. In certain cells (e.g. PtK2 cells) (*Aist et al., 1993*), cortical pulling on astral microtubules is thought to drive spindle elongation during anaphase B. However, in most biological systems studied so far (*Yu et al., 2019*; *Fu et al., 2009*; *Vukušić and Tolić, 2021*; *Brust-Mascher and Scholey, 2002*; *Vukušić et al., 2021*), this is mainly driven by molecular motors, which slide interpolar microtubules at the midzone, the central spindle region where interpolar microtubules coming from opposite poles are crosslinked antiparallelly by members of the PRC1/Ase1 family (*Figure 1A*; *Kapitein et al.,*

**Figure 1.** Characterization of microtubule dynamics during *S. pombe* anaphase B. (**A**) The three phases of mitosis in the *S. pombe* spindle. Prophase (**I**), metaphase/anaphase A (**II**) and anaphase B (**III**). Names of proteins in parenthesis indicate the markers used to label the different components. (**B**) Time-lapse images of a mitotic spindle in a cell expressing Alp7-3xGFP and Sid4-GFP. Mitotic phases are indicated on the left. Time between images is 3 min, scalebar 3 μm. (**C**) Kymograph of a mitotic spindle during anaphase in a cell expressing Alp7-3xGFP and Sid4-GFP. Time is in the vertical axis (scalebar 5 min), and space is in the horizontal axis (scalebar 2 μm). Empty arrowhead marks a growth event mentioned in the main text. Filled arrowhead marks a microtubule growth event in which the start and finish cannot be clearly determined. (**D**) Elements that can be identified in a kymograph (left) and the derived measurements (right). Pink plus ends have their minus end at the pink pole, and blue plus ends have their minus end at the blue pole. (**E–F**) Microtubule growth speed as a function of spindle length (**E**) or distance from the plus-end to closest pole (**F**) at rescue (or first point if rescue could not be exactly determined). Microtubule growth events of clear duration are shown as round blue dots, others as orange stars. Thick black lines represent average of binned data, error bars represent 95% confidence interval of the mean. Pink line in (**F**) represents a fit of the data to an error function. (**G**) Histogram showing the distribution of the position of rescues with respect to the spindle center in cells expressing Sid4-GFP and Alp7-3xGFP (blue), and position of midzone edges with respect to the spindle center in cells expressing mCherry-Atb2 and Cls1-3xGFP (green, see *Figure 1—figure supplement 1F, G*). Dashed lines represent spindle center. Cartoons below the axis in (**E–G**) illustrate how the magnitudes represented are measured. Data shown in blue and orange in (**E–G**) comes from 1671 growth events (119 cells), from 14 independent experiments (wild-type data in *Figure 2G and K* and *Figure 2—figure supplement 2E* combined). Data shown in green in (**G**) comes from 832 midzone length measurements during anaphase, from 60 cells in 10 independent experiments.

The online version of this article includes the following figure supplement(s) for figure 1:

**Figure supplement 1.** Extra wild-type characterisation.

**Figure supplement 2.** Microtubule dynamics in cells expressing GFP-Mal3.

*2008*; *Loiodice et al., 2005*; *Mollinari et al., 2002*). Importantly, sliding shortens the overlap between microtubules, such that microtubules must continuously elongate to sustain sliding (*Scholey et al., 2016*; *Vukušić and Tolić, 2021*). Remarkably, the midzone length remains roughly constant during anaphase B (*Hu et al., 2011*; *Fu et al., 2009*), indicating that net polymerisation and sliding closely match in vivo. How this coordination occurs is still poorly understood. Pioneering studies found that purified algae spindles would slide if tubulin was available for microtubules to grow, but would stop sliding in the absence of soluble tubulin (*Masuda et al., 1988*). This showed that sliding can be limited by microtubule growth. In animal cells, kinesin-4 suppresses the dynamics of interpolar microtubules and could induce this growth-limited regime (*Bieling et al., 2010*; *Hu et al., 2011*; *Hannabuss et al., 2019*). However, depletion of kinesin-4, which leads to highly dynamic interpolar microtubules (*Hu et al., 2011*), does not have an impact on anaphase spindle elongation velocity (*Vukušić et al., 2021*). Yeasts lack kinesin-4, but their final spindle length is reduced if microtubule dynamics are suppressed by nocodazole treatment or deletion of the microtubule polymerase Stu2 (*Rizk et al., 2014*), indicating that suppressing microtubule dynamics can arrest sliding. Comparable results have been obtained in HeLa cells when depleting TACC3 (*Lioutas and Vernos, 2013*). Therefore, growth-limited sliding can happen in a wide range of organisms when microtubule dynamics are suppressed, but it is not known whether it occurs in unperturbed spindles. An alternative mechanism for the coordination of sliding and growth has been recently proposed: a motor could set both the polymerisation and sliding speed of interpolar microtubules (*Krüger et al., 2021*). In fission yeast, this would be Klp9 (kinesin-6), which slides microtubules during anaphase (*Fu et al., 2009*) and promotes microtubule polymerisation in monopolar spindles (*Krüger et al., 2021*). Klp9 and the related kinesin-5 also regulate microtubule dynamics in vitro (*Chen and Hancock, 2015*; *Krüger et al., 2021*).

The contributions of microtubule nucleation and rescue to coordination of sliding and growth are less understood. Microtubule nucleation could enable sustained sliding by continuously replenishing microtubules in the overlap, similarly to what happens in *Xenopus* metaphase (*Brugués et al., 2012*). However, chromosome segregation during anaphase can occur in the absence of nucleation (*Julian et al., 1993*; *Uehara et al., 2009*; *Uehara and Goshima, 2010*). On the other hand, microtubule rescue is required during anaphase in *Xenopus* and *S. pombe*, and in the absence of the rescue promoting factor CLASP, spindle microtubules fully depolymerise at anaphase onset (*Hannak and Heald, 2006*; *Bratman and Chang, 2007*).

Finally, yeasts undergo closed mitosis: their nuclear membrane does not disassemble during mitosis, and in anaphase B it constricts to form a dumbbell shape with a thin nuclear bridge around the spindle (*Dey et al., 2020*; *Figure 1A*). At the end of anaphase, the nuclear membrane disassembles locally at the bridge, exposing the spindle to cytoplasmic factors that trigger spindle disassembly (*Lucena et al., 2015*; *Dey et al., 2020*). Les1, an inner nuclear membrane protein, forms two stalks at the sides of the bridge that restrict nuclear membrane disassembly to the spindle center and seal the edges of the severed bridge preventing leakage of nuclear content (*Dey et al., 2020*). Recent studies have highlighted the interplay between the nuclear membrane and the central spindle (*Dey et al., 2020*; *Lucena et al., 2015*; *Expósito-Serrano et al., 2020*). For example, Ase1 midzone crosslinkers are required for the sorting of nuclear pore complexes in the nuclear membrane bridge (*Expósito-Serrano et al., 2020*), and for the nuclear bridge to act as a diffusion barrier between daughter nuclei (*Boettcher et al., 2012*). Such a cross-talk between the nuclear membrane and the central spindle might affect microtubule dynamics.

Few studies have directly measured microtubule dynamics in vivo during anaphase B, which limits our understanding of how microtubule sliding and polymerisation are coordinated. FRAP has been used to infer certain aspects in cells expressing fluorescent tubulin, establishing that microtubule turnover during anaphase B is lower than during metaphase in *Drosophila* and yeast (*Brust-Mascher et al., 2004*; *Brust-Mascher et al., 2015*; *Mallavarapu et al., 1999*; *Higuchi and Uhlmann, 2005*). These experiments also showed that in yeast no microtubule nucleation occurs during anaphase B, and that interpolar microtubules are maintained through rescues (*Khodjakov et al., 2004*). Fluorescently labelled tubulin was used in yeast to image microtubule dynamics during anaphase B, but growth events were only resolvable at very late anaphase, when the spindle is composed of approximately four microtubules (*Sagolla et al., 2003*). Finally, a recent report used RPE-1 cells expressing labelled EB1, a protein that binds to the tips of growing microtubules (*Busch and Brunner, 2004*), and found that microtubule growth speed decreases with anaphase B progression (*Asthana et al.,*

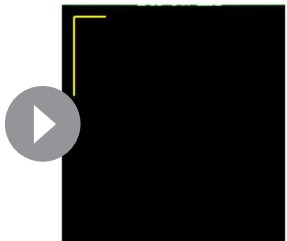
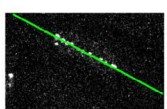

**Video 1.** Kymograph construction in cells expressing Alp7-3xGFP Sid4-GFP. Construction of the kymograph shown in Figure 1C from a live-imaging movie of cells expressing Sid4-GFP and Alp7-3xGFP. The green curve in the movie marks the fitted spindle trace (a second order polynomial) used to obtain a linear intensity profile and produce the kymograph shown on the right. In the kymograph, time is in the vertical axis (scalebar 5 min), and space is in the horizontal axis (scalebar 2 μm). The time on the top left movie is in minutes:seconds, scalebar in the movie is 2 μm.
https://elifesciences.org/articles/72630/figures#video1

2021). To our knowledge, no study to date has combined direct measurements and genetics to understand the effects of motors and MAPs on microtubule dynamics during anaphase B.

In this study, we have measured microtubule dynamics during anaphase B in *S. pombe* cells. We found that microtubule growth speed decreases when the nuclear membrane wraps around the spindle midzone. Our observations support a model in which coordination of microtubule growth and sliding is not based on a precise regulation of microtubule growth speed, but instead relies on promoting rescues at the midzone edges. This makes microtubules stable from pole to midzone, as only their distal parts including the plus ends alternate between assembly and disassembly. Consequently, the midzone persists throughout anaphase, enabling sustained sliding.

## Results
### Live-imaging of *S. pombe* cells expressing Alp7-3xGFP allows to visualise microtubule dynamics during anaphase B

To measure microtubule dynamics during anaphase, we constructed *S. pombe* strains expressing Alp7 tagged with 3xGFP at the C-terminus from its endogenous locus. Alp7 is the *S. pombe* orthologue of mammalian Transforming Acidic Coiled-Coil (TACC) (*Sato et al., 2004*), and forms a complex with the XMAP215 orthologue Alp14, a microtubule polymerase (*Al-Bassam et al., 2012*). The Alp7/Alp14 complex localises to Spindle Pole Bodies (SPBs) and microtubule plus ends (*Sato et al., 2004*). Additionally, our strains expressed the SPB marker Sid4-GFP (*Figure 1A and B*). Live-imaging of these cells during anaphase B using Structured Illumination Microscopy (SIM) produced movies where microtubule plus ends could be resolved (*Video 1*). From such movies, we constructed kymographs where the duration and velocity of microtubule growth and shrinkage events can be measured (*Figure 1C and D*). Given that in *S. pombe* all the minus ends of interpolar microtubules are located at the SPBs (*Ward et al., 2014*; *Ding et al., 1993*), microtubule length can be measured as the distance between the plus end and the pole located opposite to the direction of growth (*Figure 1D*). Note that vertical comets in kymographs (*Figure 1C*) do not correspond to non-growing microtubules, but rather to microtubules that grow roughly at the sliding speed: their minus ends are at the SPBs, which move away from the plus end in the kymograph. Finally, some comets superimpose, so not all growth events could be resolved, and we could not count the total number of microtubules. Additionally, for some growth events it was not possible to determine exactly where rescue or catastrophe happened, but the positions of SPBs and microtubule tip could be determined without ambiguity, and thus the growing speed could be measured reliably. Such events were more common at late stages of anaphase B (*Figure 1C*, filled arrowhead).

### Microtubule growth velocity decreases with anaphase B progression

We then proceeded to measure the parameters of microtubule dynamics from kymographs. We used spindle length as a proxy for anaphase B progression, as spindle elongation is very stereotypical in *S. pombe* (*Figure 1—figure supplement 1A*). Three phases can be observed corresponding to prophase (I), metaphase/anaphase A (II), and anaphase B (III) (*Figure 1A and B*, *Figure 1—figure supplement 1A*). We found that the duration of microtubule growth events did not change during anaphase B and was on average 52 ± 23 s (*Figure 1—figure supplement 1B, C*). Microtubule shrinking speed did not change during anaphase either (*Figure 1—figure supplement 1D*), and was on average 3.55 ± 1.62 μm/min (*Figure 1—figure supplement 1E*). On the other hand, microtubule growth velocity decreased during anaphase B (*Figure 1C and E*). Furthermore, this decrease was not gradual, and

we observed two populations of microtubules (fast and slow growing) characteristic of early and late anaphase. In some cells, all microtubules seemed to switch to the slow growing phase simultaneously (*Figure 1C*), while in others fast and slow growing microtubules co-existed (*Figure 2A*). A good way to visualise the two populations is to plot microtubule growth speed as a function of the distance between the plus end and the closest pole at the time of rescue (*Figure 1D and F*). This distance increases for all microtubules as the spindle elongates, and is higher for microtubules that are rescued closer to the center for a given spindle length, hence providing a reference frame aligned with the spindle. On such a plot, the data points visibly cluster in two separate clouds and the variation of growth speeds can be fitted by an error function (*Figure 1F*), which is characteristic of a system in which a transition between two states occurs at a given distance to the pole. The microtubule growth velocities of these two states, extracted from the fit, were 1.60 µm/min and 0.67 µm/min. This representation captures the fact that at early anaphase B microtubules rescued at the spindle center grow fast (empty arrowhead in *Figure 1C*), while later in anaphase fast and slow growing microtubules coexist, but those closer to the center grow slower (empty arrowheads in *Figure 2A*). Therefore, it is not anaphase progression nor position with respect to the center that best captures this behaviour, but a parameter that combines both. Consequently, fitting error functions to microtubule growth velocity vs. (1) position of rescues with respect to spindle center, (2) spindle length and (3) position of rescue with respect to the closest pole yielded $R^2$ values of 0.02, 0.39, and 0.47 respectively, indicating that position of rescue with respect to the closest pole is the best predictor of microtubule growth speed. Hence, we used this parameter for the rest of our analysis.

## Microtubule rescues occur most often at midzone edges

In our dataset, most microtubule rescues occurred near the spindle midzone. This is likely due to the fact that Cls1 (CLASP), a MAP required for microtubule rescue, is recruited to the midzone by Ase1 crosslinkers (*Bratman and Chang, 2007*). To describe the rescue distribution, we measured the positions of rescues with respect to the spindle center, considering the orientation of the microtubule to define negative and positive positions (*Figure 1D*). This revealed that rescues did not occur uniformly along the midzone, since 93% of the rescues happened at positive positions (*Figure 1G*). Moreover, the positions where most rescues happened coincided with the edge of the midzone, measured from Cls1-3xGFP signal in another strain (*Figure 1G* and *Figure 1—figure supplement 1F–G*).

## Anaphase B microtubule dynamics are similar in cells expressing GFP-Mal3 and Alp7-3xGFP

To confirm that Alp7-3xGFP can be reliably used to measure microtubule dynamics, we performed the same measurements in strains expressing GFP-Mal3 (orthologue of EB1), a protein that tracks the tips of growing microtubules, promoting microtubule growth and recruiting MAPs to the tip (*Busch and Brunner, 2004*). Cells expressing GFP-Mal3 have similar microtubule dynamics during interphase when compared to cells expressing labelled α-tubulin (*Busch and Brunner, 2004*), and GFP-tagged EB proteins are common microtubule makers (*Muroyama and Lechler, 2017*; *Rieckhoff et al., 2020*; *Reber et al., 2013*). We found that the duration of microtubule growth events was the same in cells expressing GFP-Mal3 and Alp7-3xGFP (*Figure 1—figure supplement 2A, C*), while in cells expressing GFP-Mal3, microtubule growth speed was slightly lower and rescues occurred slightly closer to the center than in cells expressing Alp7-3xGFP (*Figure 1—figure supplement 2D, E*). These differences are minor, so we concluded that Alp7-3xGFP is a reliable marker for microtubule dynamics.

Notably, GFP-Mal3 signal was lost from the spindle at late anaphase B (*Figure 1—figure supplement 2A*), making it impossible to track plus ends. This was not due to bleaching, as the GFP-Mal3 signal was not observed even if imaging was started at late anaphase B (data not shown). In contrast, Alp7-3xGFP remains at microtubule plus ends at late anaphase, making it a better marker for microtubule dynamics during anaphase. On the other hand, GFP-Mal3, unlike Alp7-3xGFP, strongly labels interphase microtubules, so we used GFP-Mal3 to compare microtubule dynamics in anaphase B and interphase (*Figure 1—figure supplement 2A, B*). We found that both microtubule growth event duration and speed were lower during anaphase B (*Figure 1—figure supplement 2F, G*). For the growth speed, we compared all interphase events with early anaphase B fast growing events (distance to the closest pole at rescue <2.5 µm, *Figure 1—figure supplement 2D*).

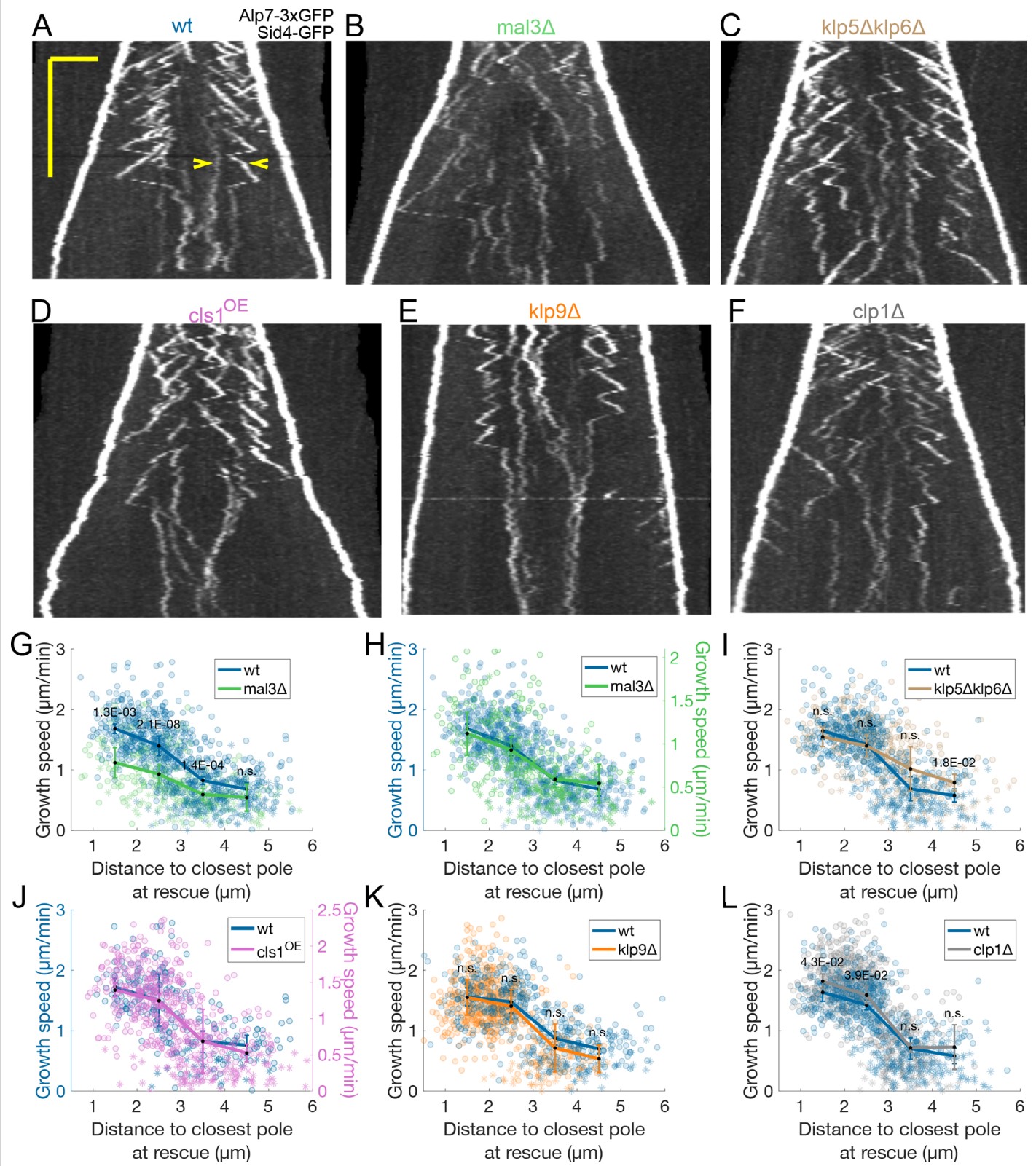

**Figure 2.** Transition from fast to slow microtubule growth occurs in the absence of important MAPs. (**A–F**) Kymographs of anaphase mitotic spindles in cells expressing Alp7-3xGFP and Sid4-GFP in different genetic backgrounds, indicated on top of each kymograph. Time is in the vertical axis (scalebar 5 minutes), and space is in the horizontal axis (scalebar 2 μm). Empty arrowheads in (**A**) mark two growth events mentioned in the main text. (**G–L**) Microtubule growth speed as a function of the distance between the plus end and the closest pole at rescue (or first point if rescue could not be

*Figure 2 continued on next page*

*Figure 2 continued*

exactly determined) in the genetic backgrounds indicated by the legends, and shown in (**A–F**). Microtubule growth events of clear duration are shown as round dots, others as stars. Thick lines represent average of binned data, error bars represent 95% confidence interval of the mean. p-values represent statistical significance of the difference of means between the two conditions at each bin (see Materials and methods). 'n.s' (not significant) indicates p > 0.05. In (**H**) and (**J**), the vertical axis for the non wild-type condition (shown on the right) is scaled. This compensates for an overall decrease in growth speed and is equivalent to a plot where the growth speed is normalised, since both vertical axes start at zero. Number of microtubule growth events shown: (**G–H**) 836 (59 cells) wt, 301 (37 cells) mal3Δ from six experiments (**I**) 589 (47 cells) wt, 424 (31 cells) klp5Δklp6Δ from five experiments (**J**) 248 (22 cells) wt, 532 (40 cells) from cls1$^{OE}$ from three experiments (**K**) 531 (36 cells) wt, 610 (40 cells) klp9Δ from four experiments (**L**) 589 (47 cells) wt, 753 (58 cells) clp1Δ from five experiments.

The online version of this article includes the following figure supplement(s) for figure 2:

**Figure supplement 1.** Comparison of growth event duration in wild-type and klp5Δklp6Δ cells.

**Figure supplement 2.** Cls1-3xGFP quantifications and extra analysis of cls1$^{off}$ and cls1$^{OE}$.

**Figure supplement 3.** Extra analysis of klp9Δ and dis1Δ cells.

**Figure supplement 4.** Rescues are clustered when deleting these MAPs.

In summary, our novel imaging approach shows that microtubule dynamics in anaphase and interphase are different. Additionally, at mid-anaphase microtubules transition from a fast growing state to a slow growing state. Finally, microtubule rescues happen with the highest probability at the midzone edge.

## Transition from fast to slow microtubule growth occurs in the absence of important MAPs

To investigate this intriguing transition from fast to slow microtubule growth velocity seen in anaphase B, we next examined various candidate MAPs associated with spindle microtubules that have been shown or proposed to regulate microtubule dynamics during anaphase B.

Bim1 (Orthologue of Mal3/EB1 in *S. cerevisiae*) is phosphorylated at late anaphase by Aurora B, which reduces its affinity for microtubules (*Zimniak et al., 2009*). In addition, Bim1 mutants that cannot be phosphorylated delay spindle disassembly (*Zimniak et al., 2009*). In *S. pombe*, Mal3 is also phosphorylated in a cell-cycle-dependent manner (*Iimori et al., 2012*), and mutant Mal3 constructs mimicking this phosphorylation have lower microtubule growth promoting activity in vitro than wild-type Mal3 (*Iimori et al., 2012*). This, combined with our observation that GFP-Mal3 leaves the spindle at late anaphase B (*Figure 1—figure supplement 2A*) made it plausible that a combination of phosphorylation and unbinding of Mal3 could lead to the decrease in microtubule growth speed. To test this possibility, we used cells deleted for mal3. As expected, considering that Mal3 promotes microtubule growth (*Busch and Brunner, 2004*), mal3Δ cells exhibited lower microtubule growth speed throughout anaphase B (*Figure 2B and G*). However, a reduction of microtubule growth could still be observed with anaphase progression, and normalised microtubule growth velocities were distributed similarly in wild-type and mal3Δ cells (*Figure 2H*), suggesting that Mal3 is not required for the reduction of microtubule growth speed in anaphase B.

*S. cerevisiae* Kip3 (kinesin-8) is a plus end directed motor that localises to the anaphase B spindle and suppresses microtubule dynamics in a length-dependent manner (*Gardner et al., 2011*; *Rizk et al., 2014*). In *S. pombe* interphase, the kinesin-8 heterodimer Klp5/Klp6 accumulates at plus ends in a length-dependent manner, promoting catastrophe (*Tischer et al., 2009*), so Klp5/Klp6 could trigger the transition from fast to slow microtubule growth during anaphase. We tested this hypothesis by deleting klp5 and klp6. As expected, klp5Δklp6Δ cells exhibited slightly longer microtubule growth events (*Figure 2—figure supplement 1*). However, the distribution of microtubule growth speed as a function of distance from the plus end to the closest pole at rescue was not very different from wild-type cells (*Figure 2C, I*), indicating that the decrease in microtubule growth speed is independent of Klp5/Klp6.

We next tested the human CLASP orthologue Cls1, a MAP that is recruited to the spindle midzone by the crosslinker Ase1. Cls1 is required for microtubule rescues to occur during anaphase in *S. pombe* (*Bratman and Chang, 2007*), and it decreases microtubule growth speed in a dose-dependent manner (*Bratman and Chang, 2007*; *Al-Bassam et al., 2010*). We measured Cls1 levels on the spindle and found that they remained constant throughout anaphase B (*Figure 2—figure*

*supplement 2A*). However, as the number of microtubules in the spindle decreases with anaphase B progression (*Ward et al., 2014*; *Ding et al., 1993*), the density of Cls1 on microtubules increases (*Figure 2—figure supplement 2B*), and this might reduce growth speed. Cls1 is an essential gene and cannot be deleted (*Grallert et al., 2006*). To study its potential role in regulating microtubule growth, we altered the levels of Cls1 by placing the gene under the control of a P81nmt1 promoter (cls1^off), which reduced the amount of Cls1 on the spindle by approximately 60% (*Figure 2—figure supplement 2A, B*), or a P1nmt1 promoter (cls1^OE), which led to a more than fourfold increase in the amount of Cls1 on the spindle (*Figure 2—figure supplement 2C, D*). Consistent with its growth suppression activity (*Bratman and Chang, 2007*; *Al-Bassam et al., 2010*), reducing Cls1 levels slightly increased microtubule growth speed, and overexpression slightly reduced it (*Figure 2—figure supplement 2E, F*). However, these changes in microtubule growth speed were minor compared to the differences on Cls1 density (*Figure 2—figure supplement 2B, D*). Additionally, as in mal3Δ cells, the normalised growth speeds were distributed similarly to wild-type in both cls1^off and cls1^OE cells (*Figure 2J*, *Figure 2—figure supplement 2G*). In other words, the transition of microtubule growth speed occurred across a wide range of Cls1 levels, arguing against a role of Cls1 in this transition.

Finally, we tested the role of Klp9 (kinesin-6), the main driver of microtubule sliding during anaphase B (*Fu et al., 2009*). Klp9 promotes microtubule growth in vitro and is required for the elongation of bundles of parallel microtubules present in monopolar spindles that undergo metaphase to anaphase transition (*Krüger et al., 2021*). Notably, the bundle elongation velocity matches the microtubule sliding speed in bipolar spindles, suggesting that Klp9 might set both microtubule growth and sliding speed in bipolar spindles. To test this hypothesis, we measured microtubule growth velocity during anaphase in cells deleted for klp9. As expected (*Fu et al., 2009*), klp9 deletion reduced spindle elongation velocity (*Figure 2—figure supplement 3A*). Interestingly, the decrease in microtubule growth velocity was delayed in klp9Δ cells with respect to wild-type (*Figure 2—figure supplement 3B, C*), while the distribution of microtubule growth speed as a function of the distance from the plus end to the closest pole at rescue was similar (*Figure 2E and K*). This could indicate that the transition in microtubule growth velocity depends primarily on the position of the plus end of the microtubule when it is rescued. Alternatively, klp9 deletion could delay the activation of a regulatory network that decreases microtubule growth speed. In any case, Klp9 is not required for the transition in microtubule growth speed to occur.

Recruitment of Klp9 to the spindle midzone relies on the dephosphorylation of its Cdk1 phosphosites at anaphase onset (*Fu et al., 2009*). This dephosphorylation occurs through two independent pathways involving the phosphatase Clp1 (orthologue of Cdc14) (*Fu et al., 2009*; *Wolfe and Gould, 2004*; *Esteban et al., 2004*), and the XMAP215 microtubule polymerase Dis1 (*Krüger et al., 2021*). Hence, microtubule growth speed could be regulated by Dis1 itself or by downstream effectors of Dis1 or Clp1. clp1Δ and dis1Δ cells had an identical phenotype to klp9Δ cells: their spindle elongation velocity was slower during anaphase, but distribution of microtubule growth speed as a function of distance from the plus end to the closest pole at rescue was similar to wild-type (*Figure 2F and L*, *Figure 2—figure supplement 3D*), indicating that neither the dephosphorylation of Cdk1 phosphosites mediated by Clp1 nor the activity of Dis1 are required for the transition from fast to slow microtubule growth velocity to occur during anaphase B.

Finally, the positions of microtubule rescues were clustered as in wild-type across all conditions (*Figure 2—figure supplement 4*). In mal3Δ and klp5Δklp6Δ cells, the average position of rescues with respect to the spindle center was markedly higher than in wild-type (*Figure 2—figure supplement 4A, B*). For klp5Δklp6Δ this might be due to longer midzones, as in budding yeast deletion of kinesin-8 increases midzone length (*Rizk et al., 2014*). It is possible that mal3Δ cells have longer midzones, or a broader distribution of overlap lengths between microtubules. Alternatively, given that Ase1 binds to Bim1 (Mal3) in budding yeast (*Thomas et al., 2020*), it is possible that this interaction is important for restricting rescues to a narrower region. Ultimately, none of the conditions perturbed the overall organisation of rescues.

In summary, our experiments show that the transition from fast to slow microtubule growth during anaphase occurs in the absence of multiple MAPs associated with the mitotic spindle known to affect microtubule dynamics in a different context or organism. Interestingly, the distribution of microtubule growth speed as a function of the distance from the plus end to the closest pole at rescue

is maintained even in cells where spindle elongation is slower, suggesting that microtubule growth velocity during anaphase B could be affected by spatial cues associated with the spindle.

## Microtubules grow slower when they enter the nuclear membrane bridge formed at the dumbbell transition

Recent reports provide increasing evidence of a cross-talk between the spindle midzone and the nuclear membrane bridge that forms after the dumbbell transition in closed mitosis (*Figure 1A*; *Dey et al., 2020*; *Expósito-Serrano et al., 2020*; *Lucena et al., 2015*). Imaging strains expressing the nuclear membrane marker Cut11-mCherry (*Figure 3A*), we noticed that the slow growing micro-tubules were the ones inside the nuclear membrane bridge (*Figure 3B and C*). To verify this, we measured microtubule growth speed in wild-type cells, cells overexpressing Klp9 (klp9$^{OE}$), and cells expressing the cell-cycle mutant allele cdc25-22, which all have different sizes and spindle elongation dynamics (*Krüger et al., 2019*). klp9$^{OE}$ cells were longer than wild-type cells, and their spindle elon-gation velocity was higher, while cdc25-22 cells were bigger than wt and klp9$^{OE}$ cells, and displayed an intermediate spindle elongation velocity (*Figure 3—figure supplement 1A–C*). Wild-type, klp9$^{OE}$and cdc25-22 nuclei underwent dumbbell transition at different spindle lengths (*Figure 3—figure supplement 1D*), and their distribution of microtubule growth speed as a function of time, spindle length or distance from the plus end to the closest pole at rescue were different (*Figure 3—figure supplement 1E–H*). However, regression analysis showed that if we categorised microtubule growth events depending on whether they occurred 'before' the dumbbell transition, and 'inside' or 'outside' the nuclear membrane bridge (cartoons in *Figure 3D*), microtubule growth velocity in each category was not different for wt, klp9$^{OE}$ and cdc25-22 cells (*Figure 3D*). In fact, a linear model taking into account only this categorisation, explained 48% of the variability (see Statistical Analysis). The 13% decrease in growth speed observed between 'before' and 'outside' events was minor compared to the 60% reduction when comparing 'before' and 'inside' (*Figure 3D*). This categorisation clarifies our plots of microtubule growth speed as a function of the distance from the plus end to the closest pole at rescue (*Figure 1F*), as this distance is equivalent to the distance from the plus end to the nuclear membrane bridge edge (*Figure 3E*).

From this direct imaging we conclude that microtubules grow slower when they enter the nuclear membrane bridge formed by the dumbbell transition. The wild-type microtubule dynamics are summarised in *Table 1*.

## Preventing the dumbbell transition abolishes the switch from fast to slow microtubule growth

To check whether the nuclear membrane bridge is indeed causing microtubules to grow slower at mid-anaphase, we prevented the nuclear membrane dumbbell transition in two different ways.

First, we inhibited Aurora B using an analogue sensitive allele (ark1-as3) (*Hauf et al., 2007*), which often led to failed chromosome segregation (*Petersen et al., 2001*), and spindles that elongated without undergoing dumbbell transition (*Figure 4B*). To distinguish the direct effects of Aurora B inhi-bition from the effects of preventing the dumbbell transition, we added the analogue 1NM-PP1 and imaged cells immediately. Some of the cells were in anaphase B when the analogue was added, and displayed normal chromosome segregation ('ark1-as3 normal'), others failed chromosome segrega-tion and did not undergo dumbbell transition ('ark1-as3 abnormal'). In wild-type and 'ark1-as3 normal' cells, microtubules grew slower inside the nuclear membrane bridge (*Figure 4I*, *Appendix 1—table 2*), but in 'ark1-as3 abnormal' cells microtubule growth velocity was not significantly different (p-value 0.38, see Statistical Analysis) outside or inside the nuclear membrane tubes formed at the spindle poles (*Figure 4F, I Figure 4—figure supplement 1B*). This data shows that the transition from fast to slow microtubule growth can happen upon Aurora B inactivation, but not if chromosome segregation fails and the dumbbell transition is prevented.

Next, we examined cells treated with the fatty acid synthetase inhibitor cerulenin, which reduces cellular membrane availability, dramatically decreasing spindle elongation, and prevents dumbbell transition (*Figure 4C*; *Yam et al., 2011*). Treating cells with 10 µM of cerulenin prevented dumbbell transition, and abolished the decrease in microtubule growth speed (*Figure 4C and G*, *Figure 4—figure supplement 1C*). Furthermore, out of the 23 cells treated with cerulenin, we observed a clear bimodal growth velocity distribution only in two cells that underwent dumbbell transition despite

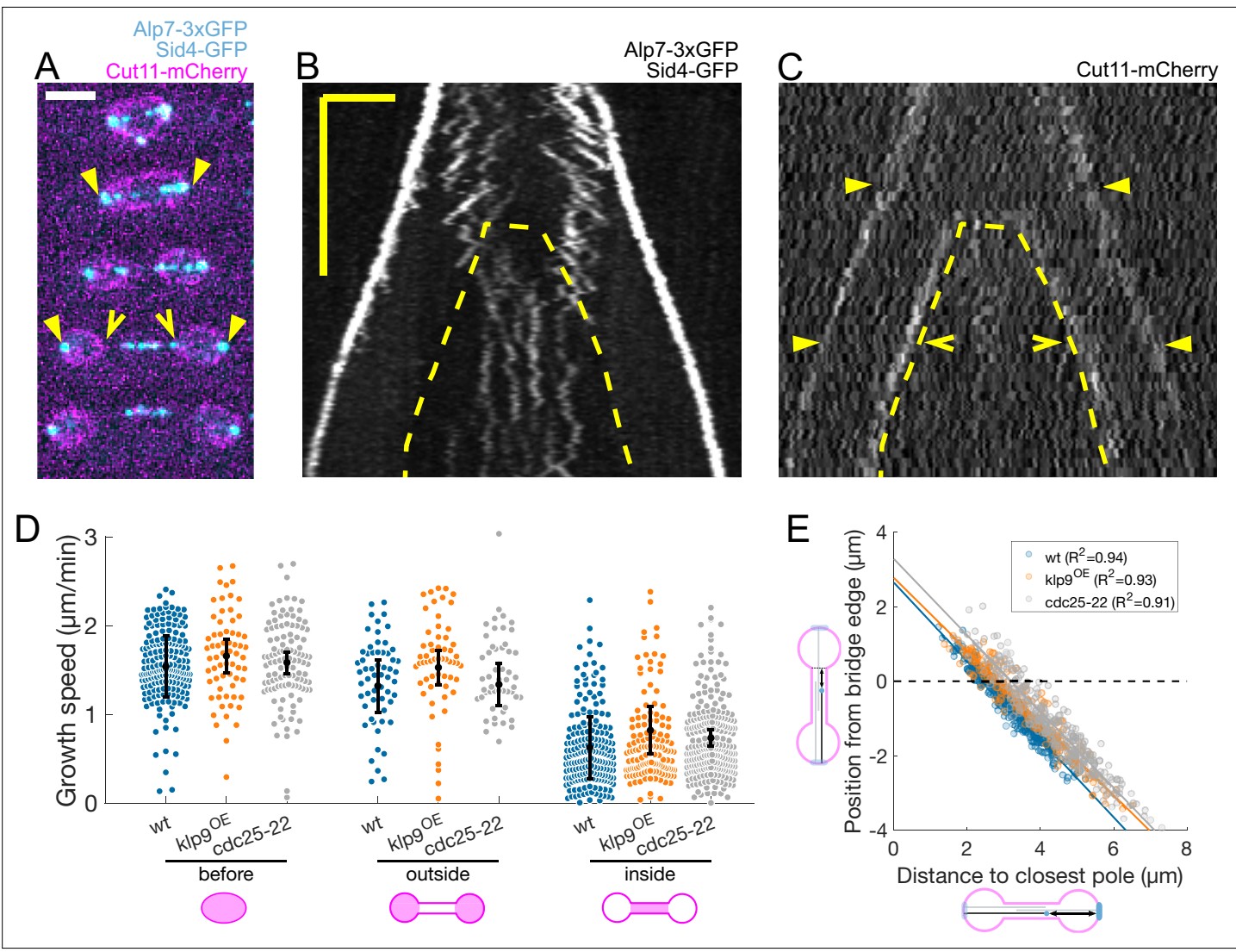

**Figure 3.** Microtubules grow slower when they enter the nuclear membrane bridge formed at the dumbbell transition. (**A**) Time-lapse images of an anaphase B mitotic spindle in a cell expressing Alp7-3xGFP, Sid4-GFP (cyan) and Cut11-mCherry (magenta). Time between images is 3 minutes, scalebar 3 µm. Filled arrowheads denote spindle poles, and empty arrowheads denote limits of the nuclear membrane bridge. Equivalent positions are marked in the kymograph in (**C**). (**B–C**) Kymographs of an anaphase B mitotic spindle in a cell expressing Alp7-3xGFP, Sid4-GFP (**B**) and Cut11-mCherry (**C**). Time is in the vertical axis (scalebar 5 min), and space is in the horizontal axis (scalebar 2 µm). Dashed lines outline the nuclear membrane bridge formed after the dumbbell transition (see *Figure 1A*). See legend of (**A**) for arrowheads. (**D**) Microtubule growth speed in wild-type (blue), klp9$^{OE}$ (orange), and cdc25-22 (grey) cells. Events are categorised according to whether rescue occurred before the dumbbell transition, and inside or outside the nuclear membrane bridge (see cartoons under x-axis). Error bars represent 95% confidence interval of the mean. For values of confidence intervals and statistical significance see *Appendix 1—table 1*. (**E**) Distance from the plus-end to the nuclear membrane bridge edge at rescue as a function of distance from the plus-end to the closest pole at rescue. Dots represent individual microtubule growth events, with colour code as in (**D**). Lines represent first-degree polynomial fit to the data in each condition, of which the R$^2$ is shown in the legend. Number of events: 442 (30 cells) wt, 260 (27 cells) klp9$^{OE}$, 401 (35 cells) cdc25-22, from three independent experiments.

The online version of this article includes the following figure supplement(s) for figure 3:

**Figure supplement 1.** Comparison of wild-type, klp9$^{OE}$ and cdc25-22 cells.

the treatment (growth events in the 'inside' and 'outside' categories in *Figure 4J*, and *Figure 4— figure supplement 1E*). By reducing cerulenin concentration to 5 µM, we observed two additional cells in which a reduction of microtubule growth speed after abrupt dumbbell transition was evident (*Figure 4H*, *Figure 4—figure supplement 1D, F*).

In summary, our data shows that microtubule growth speed can remain unchanged for a period of time as long as wild-type anaphase B if the dumbbell transition is prevented (*Figure 4F and G*). This

**Table 1.** Anaphase B microtubule dynamics. Mean and standard deviation (S.D.) for several magnitudes of microtubule dynamics measured from kymographs of cells expressing Alp7-3xGFP, Sid4-GFP, and Cut11-mCherry. The microtubule growth speed is categorised according to whether rescue occurred before the dumbbell transition, and inside or outside the nuclear membrane bridge (see cartoons in *Figure 3*). Data from cells grown in YE5S medium and imaged at 27 °C in the absence of DMSO (wild-type in *Figure 5F* and *Figure 4—figure supplement 2F*).

|  | Mean ± S.D. |
| --- | --- |
| Growth speed before | 1.54 ± 0.31 µm/min |
| Growth speed outside | 1.37 ± 0.31 µm/min |
| Growth speed inside | 0.66 ± 0.44 µm/min |
| Growth event duration | 49 ± 19 s |
|  |  |
| Position of plus-end with respect to spindle center at rescue | 1.02 ± 0.48 µm |
| Shrinkage speed | 4.12 ± 1.56 µm/min |

suggests that the decrease in microtubule growth speed observed in wild-type cells is not regulated by a 'timer' mechanism, but instead occurs when the nuclear membrane bridge encloses the midzone.

## Formation of Les1 stalks is required for normal decrease in growth speed associated with internalisation of microtubules in the nuclear membrane bridge

To further understand whether an interaction between the nuclear membrane bridge and the spindle affects microtubule growth, we studied the effect of les1 and nem1 deletions on microtubule growth during anaphase (*Figure 4—figure supplement 2A–C*). Les1 forms stalks at the edges of the nuclear membrane bridge that restrict nuclear membrane disassembly to the center of the bridge and may constitute sites of close interaction between the nuclear membrane and the spindle (*Dey et al., 2020*). nem1 deletion causes an increase in nuclear size (*Kume et al., 2019*), which leads to membrane ruffling (*Figure 4—figure supplement 2D*) and prevents the formation of Les1 stalks (*Dey et al., 2020*).

We found that in both les1Δ and nem1Δ cells microtubule growth speed inside the nuclear bridge was faster than in wild-type cells, while the microtubule growth speed outside the nuclear bridge was unaffected (*Figure 4—figure supplement 2E–F*, *Appendix 1—table 5*, *Appendix 1—table 6*). This data suggests that the formation of Les1 stalks (which potentially promote close interaction of spindle microtubules with the nuclear envelope) is required for the normal decrease of microtubule growth speed when microtubule plus ends enter the nuclear membrane bridge.

## Ase1 is required for normal rescue distribution

Preventing the decrease in microtubule growth speed by cerulenin treatment or Aurora B inhibition did not compromise spindle stability nor overall microtubule organisation. This suggested that the way microtubule sliding and growth are coordinated is robust against deviations from the normal microtubule growth speed evolution. In fact, since rescues occur most often at midzone edges (*Figure 1G*), microtubules are stable from pole to midzone, and only their distal parts including the plus ends alternate between assembly and disassembly phases (Figure 7A). Therefore, promoting rescues at the midzone edges could be sufficient for the spindle to maintain a constant midzone length while sustaining sliding. We initially set out to test this mechanism experimentally by deleting ase1, a microtubule crosslinker that organises the midzone (*Loiodice et al., 2005*; *Yamashita et al., 2005*) and recruits the rescue factor Cls1 (*Bratman and Chang, 2007*). As expected (*Loiodice et al., 2005*; *Yamashita et al., 2005*; *Bratman and Chang, 2007*), in cells where ase1 was deleted the characteristic distribution of rescues was lost (*Figure 5B and E*) and spindles collapsed due to loss of microtubule overlaps before reaching the typical final length (*Figure 5B–D Figure 5—figure supplement 1*).

## The decrease in growth speed associated with internalisation of microtubules in the nuclear membrane bridge is reduced upon Ase1 deletion

We next examined the microtubule growth speed in ase1Δ cells. Approximately half of ase1Δ cells (30 out of 63) did not reach sufficiently long spindle lengths to undergo dumbbell transition, but we restricted our analysis to ase1Δ cells that did undergo dumbbell transition. Interestingly, we observed

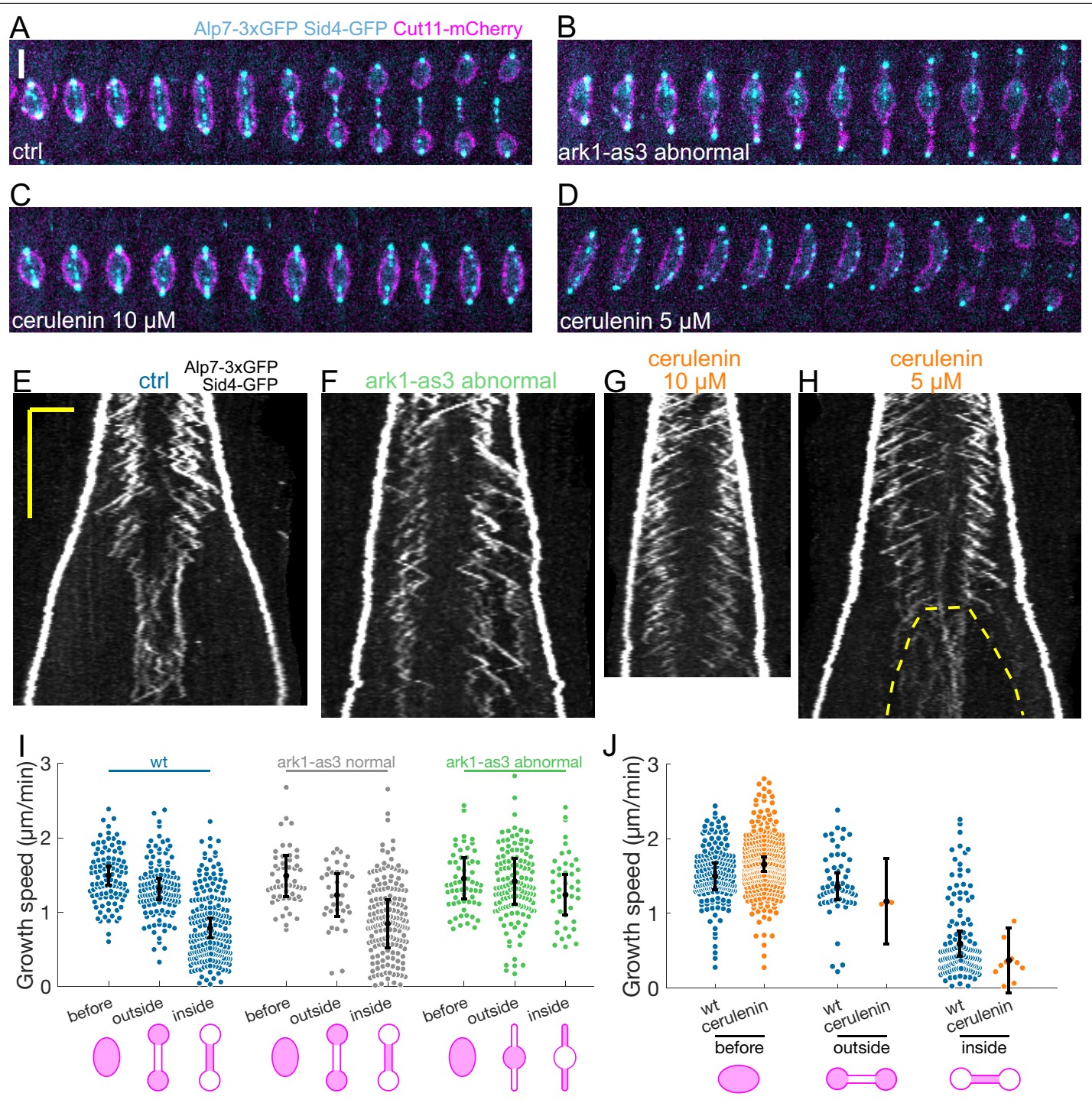

**Figure 4.** Preventing the dumbbell transition abolishes the switch from fast to slow microtubule growth. (A–D) Time-lapse images of mitotic spindles in cells expressing Alp7-3xGFP, Sid4-GFP (cyan) and Cut11-mCherry (magenta), in different conditions: (A) wild-type +DMSO (cerulenin control), (B) ark1-as3 with 5 µM 1NM-PP1, (C) wild-type with 10 µM cerulenin, (D) wild-type with 5 µM cerulenin in which the nuclear membrane eventually undergoes dumbbell transition. Time between images is 1 min, scalebar 3 µm. (E–H) Kymographs of anaphase mitotic spindles in cells expressing Alp7-3xGFP, Sid4-GFP and Cut11-mCherry in the same conditions as (A–D). Time is in the vertical axis (scalebar 5 min), and space is in the horizontal axis (scalebar 2 µm). Dashed line in (H) outlines the nuclear membrane bridge formed after the dumbbell transition. See corresponding Cut11-mCherry kymographs in *Figure 4—figure supplement 1A–D*. (I) Microtubule growth speed in cells treated with 5 µM 1NM-PP1, blue: wild-type cells, grey: ark1-as3 cells that underwent normal chromosome segregation ('ark1-as3 normal'), green: ark1-as3 cells that failed chromosome segregation and did not undergo dumbbell transition ('ark1-as3 abnormal'). Events are categorised according to where the rescue occurred (see cartoons under x-axis. Note that for

*Figure 4 continued on next page*

*Figure 4 continued*

'ark1-as3 abnormal', the meaning of 'outside' and 'inside' categories is different due to the failure to undergo dumbbell transition). (**J**) Microtubule growth speed in wild-type cells treated with DMSO (blue) or 10 µM cerulenin (orange). Events are categorised according to where the rescue occurred (see cartoons below). Error bars represent 95% confidence interval of the mean. For values of confidence intervals and statistical significance see *Appendix 1—Tables 2 and 3*. Number of events: (**I**) 446 (25 cells) wt, 257 (20 kymographs) ark1-as3 normal, 240 (17 cells) ark1-as3 abnormal, from four experiments. (**J**) 368 (28 cells) wt, 328 (23 cells) cerulenin, from four experiments.

The online version of this article includes the following figure supplement(s) for figure 4:

**Figure supplement 1.** Kymographs of the Cut11-mCherry channel and more cerulenin examples.

**Figure supplement 2.** Formation of Les1 stalks is required for normal decrease in growth speed associated with internalisation of microtubules in the nuclear membrane bridge.

that the microtubule growth speed was higher throughout anaphase in ase1Δ when compared to wild-type, but also that the decrease in microtubule growth between 'before' and 'inside' categories was ~25%, half of the ~50% decrease observed in wild-type cells (*Figure 5F*, *Appendix 1—table 4*). Plotting the microtubule growth speed as a function of distance to the closest pole at rescue (*Figure 5G*) showed that this effect was stronger that the conditions tested in *Figure 2*.

To better understand the effect of Ase1 on microtubule growth speed, we compared strains expressing mCherry-ase1 from the ase1 endogenous promoter or P1nmt1 (mCherry-ase1^OE^), which resulted in a more than 10-fold increase of mCherry-Ase1 levels on the spindle (*Figure 5—figure supplement 2A–F*). Microtubule growth speed in mCherry-ase1^OE^ cells was similar to wild-type at early anaphase, but decreased more sharply with distance to the pole (*Figure 5—figure supplement 2G*). We next reduced mCherry-ase1 levels using a P81nmt1 promoter, but the signal was undistinguishable from noise (data not shown). As an alternative, we quantified the levels of GFP-Ase1 in strains expressing GFP-ase1 either from its endogenous promoter or P81nmt1 (GFP-ase1 ^off^). GFP-Ase1 levels on the spindle were approximately threefold lower in GFP-ase1 ^off^ cells (*Figure 5—figure supplement 3C, D*). We then measured microtubule growth velocity in cells expressing unlabelled ase1 from a P81nmt1 promoter (ase1^off^, *Figure 5—figure supplement 3B*). ase1^off^ cells exhibited a slight increase in microtubule growth (*Figure 5—figure supplement 3E*) throughout anaphase, but normalised growth speed as a function of distance from the plus end to the closest pole at rescue was similar to wild-type (*Figure 5—figure supplement 3F*). These results are reminiscent of those obtained when altering Cls1 levels (*Figure 2*, *Figure 2—figure supplement 2*), so they may be partially due to altered recruitment of Cls1 by Ase1. However, like for Cls1, we observed a similar ratio between growth speed at early and late anaphase across a wide range of Ase1 levels (10x to 1/3), arguing against a direct effect of Ase1 on microtubule growth. The strong effects in ase1Δ may indicate that Ase1 is required for the nuclear membrane bridge to have an effect on microtubule growth (see Discussion). Conversely, we cannot rule out that deletion or complete inactivation of cls1 may have a comparable effect to deletion of ase1, but this cannot be tested, as it leads to the immediate disassembly of the spindle at anaphase onset (*Bratman and Chang, 2007*).

In summary, Ase1 is required for rescue organisation and for microtubule growth speed to decrease normally during anaphase B.

## Promoting microtubule rescues at the midzone edge is sufficient to coordinate sliding and growth across a wide range of microtubule growth speeds

Our observations showed that anaphase B spindles were robust against perturbations on microtubule growth (*Figure 4*), but sensitive to rescue distribution (*Figure 5*). Therefore, we asked whether restricting rescues to the midzone edges is sufficient to coordinate microtubule sliding and growth. We developed a minimal mathematical model (see Materials and methods) to compare scenarios that differ in how rescue is promoted inside the midzone. In our model, the spindle is initially composed of nine antiparallel microtubules (*Figure 6A*) that undergo dynamic instability. For simplicity, we do not specifically simulate motors and MAPs, but we assume a constant spindle elongation speed and define the midzone as a region of the space of fixed length. Based on the fact that Cls1 is recruited by Ase1 (*Bratman and Chang, 2007*) and that Cls1 levels remain constant during anaphase (*Figure 2—figure supplement 2A*), we assume that a fixed amount of rescue factor (*R*) is distributed along

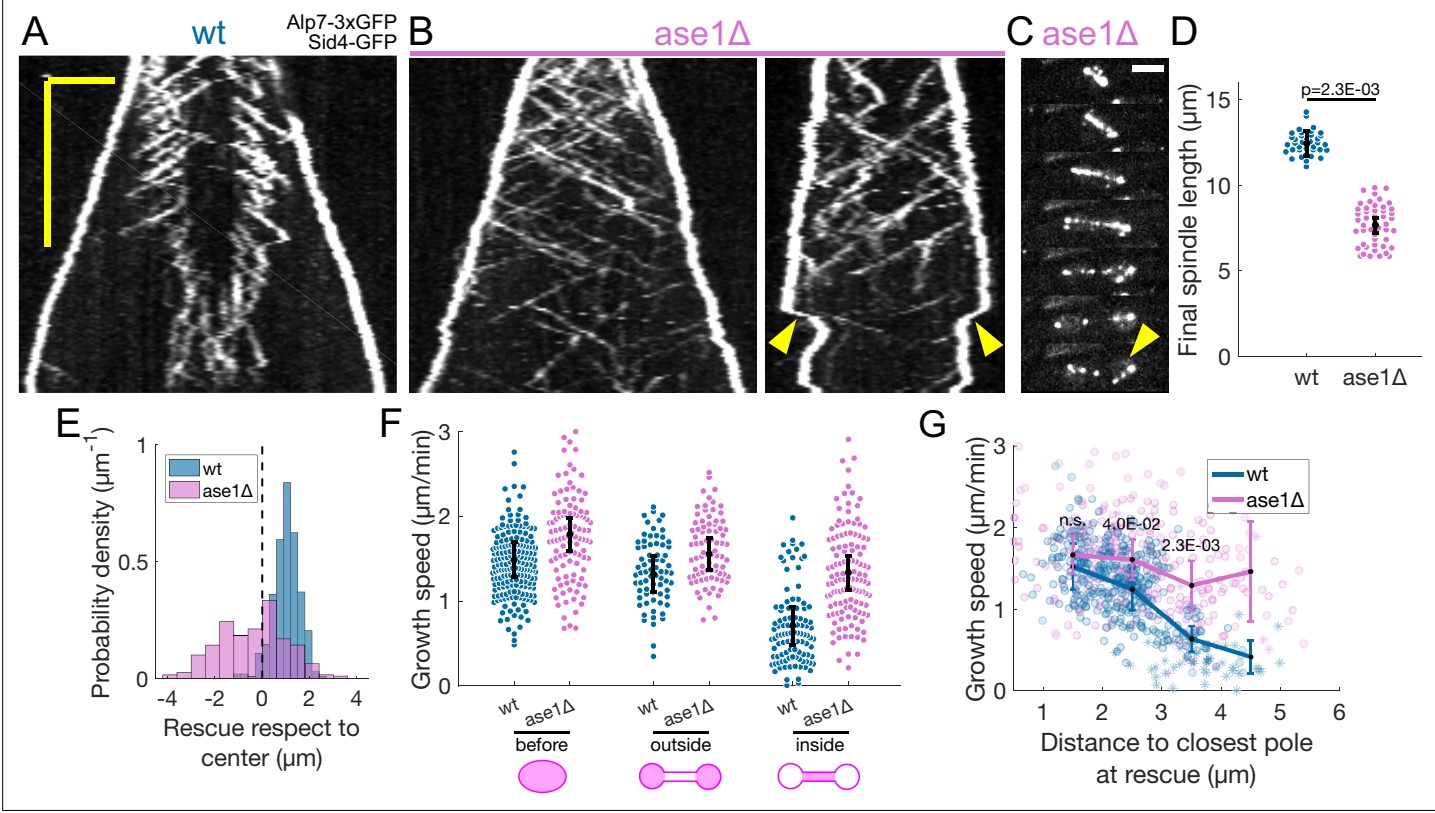

**Figure 5.** Ase1 is required for normal rescue distribution and for microtubule growth speed to decrease normally during anaphase B. (**A, B**) Kymographs of anaphase mitotic spindles in wild-type (**A**) and ase1Δ (**B**) cells expressing Alp7-3xGFP and Sid4-GFP. Time is in the vertical axis (scalebar 5 min), and space is in the horizontal axis (scalebar 2 μm). (**C**) Time-lapse images of a mitotic spindle in a cell where ase1 is deleted, expressing Alp7-3xGFP and Sid4-GFP. Time between images is 3 min, scalebar 3 μm. Arrowhead indicates spindle collapse that occurs due to loss of microtubule overlap. (**D**) Distribution of final spindle length in wild-type and ase1Δ cells. Same data as in *Figure 5—figure supplement 1*. Error bars represent 95% confidence interval of the mean, p-value represents statistical significance of the difference of means between the two conditions (see Materials and methods). (**E**) Distribution of the position of rescues with respect to the spindle center in wild-type and ase1Δ cells. Dotted line marks the spindle center. (**F**) Microtubule growth speed in wild-type (blue) and ase1Δ (pink) cells. Events are categorised according to whether rescue occurred before the dumbbell transition, and inside or outside the nuclear membrane bridge (see cartoons under x-axis). Error bars represent 95% confidence interval of the mean. For values of confidence intervals and statistical significance see *Appendix 1—table 4*. (**G**) Microtubule growth speed as a function of the distance between the plus end and the closest pole at rescue (or first point if rescue could not be exactly determined) in wild-type (blue) and ase1Δ (pink) cells expressing Sid4-GFP Alp7-3xGFP and Cut11-mCherry. Microtubule growth events of clear duration are shown as round dots, others as stars. Thick lines represent average of binned data, error bars represent 95% confidence interval of the mean. p-values represent statistical significance of the difference of means between the two conditions at each bin (see Materials and methods). 'n.s' (not significant) indicates p > 0.05. Number of events shown: (**D**) 37 wt, 48 ase1Δ cells from three independent experiments. (**E**) 402 (34 cells) wt, 316 (39 cells) ase1Δ microtubule growth events from four independent experiments. (**F, G**) 356 (24 cells) wt, 310 (35 cells) ase1Δ microtubule growth events from 6 independent experiments. Only the cells which underwent dumbbell transition were used for the analysis in (**F, G**).

The online version of this article includes the following figure supplement(s) for figure 5:

**Figure supplement 1.** ase1Δ spindle elongation dynamics.

**Figure supplement 2.** Overexpression of mCherry-ase1.

**Figure supplement 3.** Shutoff of GFP-ase1 and ase1.

the overlap between microtubules inside the midzone. The rescue activity is either uniform along the midzone, or distributed according to a beta distribution with parameters $\alpha$ and $\beta$. Increasing $\alpha$ localises the rescue activity at the midzone edges to higher degrees, while keeping the total rescue factor constant (*Figure 6A*). All the model parameters were derived from experimental measurements, except for $R$, $\alpha$ and $\beta$ (*Table 2*). $R$ was scanned and tuned to fit the experiments (see below). We explored $\alpha=\beta=1$, corresponding to a uniform distribution, and $\alpha = 4, 8, 12$ with $\beta=2$, representing increasingly skewed distributions towards the midzone edge (*Figure 6A*). We initially set the microtubule growth velocity to 1.6 μm/min (early anaphase speed, *Figure 1F*), and aimed to reproduce

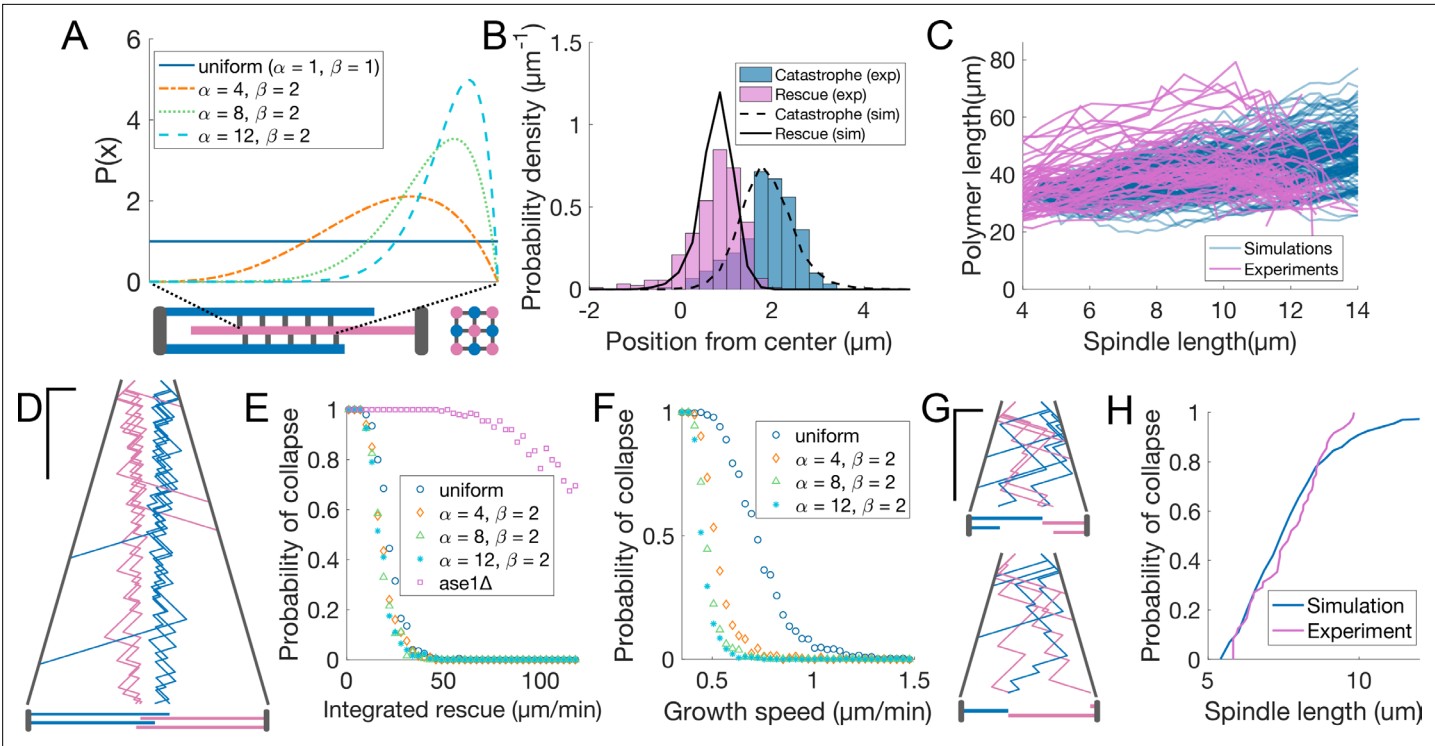

**Figure 6.** Promoting microtubule rescues at the midzone edge is sufficient to coordinate sliding and growth. (**A**) Arrangement of microtubules in simulations, shown as longitudinal (bottom left) and perpendicular (bottom right) sections of the spindle. Microtubules are colour coded according to their orientation, SPBs and connections between microtubules are shown in grey. The dashed lines linking the midzone edges to the extremes of the x-axis represent the fact that the parameter $x$ maps the position along the midzone to a value that goes from zero to one. Curves inside the graph represent the value of $P(x)$ from *Equation 5* (see Materials and methods) for parameters indicated in the legend. (**B**) Distribution of positions of microtubule catastrophe and rescue with respect to the spindle center in experiments (histograms, same data as *Figure 1G*), and 200 simulations (lines), for $R = 55$ μm/min, $\alpha=4$, $\beta=2$. (**C**) Total polymerised tubulin as a function of spindle length in 200 simulations (blue) and total mCherry-atb2 intensity (scaled) as a function of spindle length in 60 cells from six independent experiments (pink). Simulation Parameters as in (**B**). (**D**) Kymograph generated from a simulation with parameters as in (**B**). Plus-ends are colour coded according to their orientation, SPBs are shown in grey. Time is in the vertical axis (scalebar 5 min), and space is in the horizontal axis (scalebar 2 μm). Total simulated time is 20 min. Cartoon below the kymograph shows the lengths of microtubules in the spindle at the last timepoint in the kymograph. (**E**) Probability of spindle collapse as a function of the total rescue factor ($R$). Each dot represents a set of 200 simulations with equal parameters (*Table 2*), and its value on the y axis is the fraction of the simulations in which the spindle collapsed in the 20 min of simulated time. (**F**) Same as (**E**), but each dot represents 500 simulations, for various values of microtubule growth speed. (**G**) Simulation kymograph as in (**D**), for ase1Δ, where rescue activity is uniformly distributed along the whole length of microtubules. Both simulations ended with a spindle collapse due to loss of the overlap between antiparallel microtubules (see cartoons below). $R = 34$ μm/min (**H**) Cumulative distribution of final spindle length in ase1Δ cells (pink, same data as *Figure 5E*), and in 200 ase1Δ simulations (blue). See *Table 2* for simulation parameters.

The online version of this article includes the following figure supplement(s) for figure 6:

**Figure supplement 1.** Additional model figures.

the experimental distribution of positions of rescue and catastrophe at early anaphase (spindle length <6 μm, *Figure 6B*), measured from kymographs (*Figure 1C*), and the total tubulin intensity as a function of spindle length (*Figure 6C*), measured in cells expressing fluorescent α-tubulin (*Figure 1—figure supplement 1F*). Fluorescent tubulin intensity is proportional to polymerised tubulin (*Ward et al., 2014*; *Loiodice et al., 2019*), so it can be compared to the total polymerised tubulin in a simulation by using a scaling factor (*Figure 6C*, see Materials and methods).

A good agreement with the experimental data was obtained with $\alpha=4$, $R=55$ μm/min (*Figure 6B–D*, *Figure 6—figure supplement 1D*). For higher values of $R$, the spindle was stable (*Figure 6E*) but, due to the higher microtubule stability, the total polymerised tubulin increased steadily, unlike in the experiment, where it remained more or less constant throughout anaphase (*Figure 6C*). Importantly, the distributions of rescues and catastrophes were similar to the experimentally observed when the

**Table 2.** Simulation parameters.

| Symbol | Meaning | Value | Source |
|---|---|---|---|
| $v_p$ | Microtubule growth speed | 1.6 μm/min when fixed, uniformly sampled between 0.35 and 1.5 μm/min when scanned | *Figure 1F* |
| $v_d$ | Microtubule shrinking speed | 3.6 μm/min | *Figure 1—figure supplement 1D* |
| $v_s$ | Microtubule sliding speed | 0.35 μm/min | *Figure 1—figure supplement 1A* |
| $R$ | Integrated rescue rate | 55 μm/min when fixed for wild-type, 34 μm/min when fixed for ase1Δ, uniformly sampled between 1 and 120 μm/min when scanned | |
| $n$ | See *Equation 10* | 8.53 in wild-type, 6.8 in ase1Δ | *Figure 6—figure supplement 1B* for wt, data not shown for ase1Δ |
| $\theta$ | See *Equation 10* | 3.17 min$^{-1}$ in wild-type, 2.5 min$^{-1}$ in ase1Δ | *Figure 6—figure supplement 1B* for wt, data not shown for ase1Δ |
| μ | Mean of normal fit | 1.23 μm | *Figure 6—figure supplement 1C* |
| $\sigma$ | Standard devitation of normal fit | 0.25 μm | *Figure 6—figure supplement 1C* |
| $\alpha$ | Parameter of beta distribution | 1 in uniform, indicated in legend otherwise | |
| $\beta$ | Parameter of beta distribution | 1 in uniform, 2 otherwise | |
| $h$ | Simulation timestep | 0.01 min | |

rescue factor was uniformly distributed along the midzone (*Figure 6—figure supplement 1E*) because even in that case, the most likely place to have a rescue is the first position where it can happen.

For microtubule growth speed characteristic of early anaphase (1.6 μm/min), the spindle stability was not affected by the distribution of rescue rate within the midzone (*Figure 6E*). In contrast, if we kept $R$ as in *Figure 6B–D*, but decreased the microtubule growth speed, we observed that spindles with skewed rescue distributions were more stable (*Figure 6F*). When growth velocity is low, a rescued microtubule might not exit the midzone before undergoing catastrophe, and therefore 'miss' a fraction of the midzone with rescue activity. Increasing the rescue rate close to the midzone edge makes it more likely that plus ends of rescued microtubules are outside the midzone by the time they undergo catastrophe, which maximises their chances to be rescued again.

In summary, this simple model shows that promoting rescues at midzone edges is sufficient to maintain the microtubule overlap and sustain sliding across a wide range of microtubule growth speeds

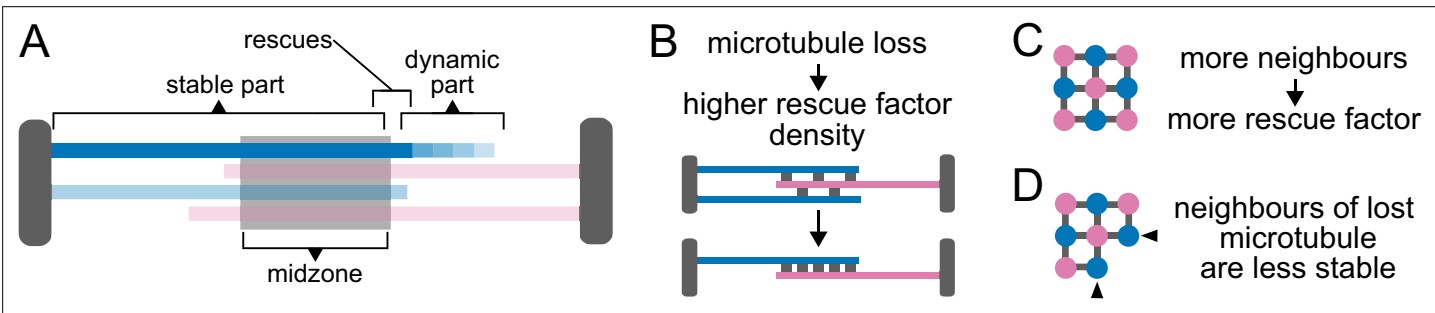

**Figure 7.** Model schematic Cartoons depicting the main consequences of the model assumptions. SPBs, the midzone and rescue factor distributed between microtubule overlaps are shown in grey. Microtubules are colour coded according to their orientation. (**A**) and (**B**) show longitudinal sections of the spindle. (**C**) and (**D**) show perpendicular sections. (**A**) Since rescues occur most often at midzone edges (*Figure 1G*), microtubules are stable from pole to midzone, and only their distal parts including the plus ends alternate between assembly and disassembly phases. (**B**) Because rescue factor levels on the spindle remain constant during anaphase (*Figure 2—figure supplement 2A*), spindles become increasingly stable as they lose microtubules. (**C**) We assume that the fixed amount of rescue factor distributes along overlaps, so rescue rate is proportional to the number of neighbours of a microtubule. (**D**) Therefore, when a microtubule depolymerises, its antiparallel neighbours lose one neighbour, while the microtubules in the same orientation as the lost one keep the same number of neighbours. This simple effect promotes even loss of microtubules from both poles.

(*Figure 7A*). Within our limited exploration, it suggests that accumulating the rescue activity at the midzone edges might represent an optimal localisation of this activity, in the context of anaphase B.

## Distributing a fixed amount of rescue factor along midzone overlaps leads to increasing microtubule stability and even loss of microtubules from both spindle poles

Because Cls1 levels on the spindle remain constant during anaphase (*Figure 2—figure supplement 2A*), our model assumes that a fixed amount of rescue factor *R* distributes along the midzone overlap, which makes spindles increasingly stable as they lose microtubules (*Figure 7B*). This mechanism might operate in cells, since anaphase spindles can last for up to 40 min if spindle disassembly is delayed (*Figure 2—figure supplement 3A*). Additionally, assuming that a fixed amount of rescue factor distributes along overlaps makes rescue rate proportional to the number of neighbours of a microtubule (*Figure 7C*). Therefore, when a microtubule fully depolymerises, its neighbours oriented in the opposite direction lose one neighbour, while the microtubules in the same orientation as the lost one keep the same number of neighbours. This simple effect promotes even loss of microtubules from both poles (*Figure 7D*), which is necessary to prevent spindle collapse during anaphase B in *S. pombe*. To show that indeed microtubules are lost evenly from both poles in cells, let us consider a representative spindle that has nine microtubules at anaphase B onset. By the end of anaphase, approximately five microtubules will remain (*Ward et al., 2014*). Thus, roughly 1/2 of the microtubules are lost. Assuming that this probability of being lost (1/2) is the same for all microtubules, for a spindle with nine microtubules at anaphase onset (four from one pole and five from the other pole), the probability of losing all microtubules from either pole by the end of anaphase would then be $\sim \frac{1}{2}^4 + \frac{1}{2}^5$, so approximately 10% of spindles would collapse because of this. In contrast, spindles do not collapse in our simulations, where a mechanism driving even loss of microtubules operates. In 500 simulations with parameters as in *Figure 6B and C*, where spindles start anaphase with nine microtubules, and have on average five microtubules after 15 min (the typical duration of anaphase B) we observed no spindle collapse (*Figure 6F*). Therefore, this simple mechanism is remarkably effective.

## Loss of microtubule rescue organisation leads to spindle collapse

Finally, we tested whether the loss of rescue organisation observed in ase1Δ cells (*Figure 5E*) could be partly responsible for spindle collapse (*Figure 5C and D*). We modified the model so that the total amount of rescue factor was distributed all along microtubules, and the rescue rate was the same anywhere on the spindle. By scanning the parameter representing the total rescue factor (*R*), we found that most spindles collapsed due to loss of antiparallel overlap of microtubules in 20 min of simulated time, even for values of *R* way higher than those required to maintain stability in spindles with midzone (*Figure 6E*). This is similar to what happens in ase1Δ cells, where interpolar microtubules coming from opposite poles often lose their connection prior to reaching the typical final spindle length (*Figure 5C and D*). For *R*=34 μm/min, this modified model closely reproduced the distribution of spindle length at collapse observed in ase1Δ cells (*Figure 6G and H*, *Figure 6—figure supplement 1F*), suggesting that in these cells a combination of lower rescue rate and loss of rescue organisation leads to spindles collapsing prior to reaching the typical final spindle length.

## Discussion

We have measured microtubule dynamics during anaphase B in *S. pombe* to examine how microtubule polymerisation and sliding are coordinated. We found that: (1) Wrapping of the nuclear membrane around the spindle midzone reduces microtubule growth speed. (2) Rescues occur preferentially at midzone edges. We then developed a model showing that organising rescues in this manner is sufficient to coordinate microtubule growth and sliding across the wide range of microtubule growth speeds observed experimentally.

## Wrapping of the nuclear membrane around the spindle midzone reduces microtubule growth speed

In animal cells, microtubule dynamics are increasingly suppressed during anaphase B by kinesin-4 (*Hu et al., 2011*; *Nunes Bastos et al., 2013*; *Asthana et al., 2021*). We have found that in *S. pombe*, which

lacks kinesin-4, microtubule growth speed also decreases during anaphase B (*Figure 1E*). Microtubules exist in two states (fast and slow growing) characteristic of early and late anaphase (*Figure 1F*). The transition between these states occurs independently of several motors and MAPs that regulate microtubule dynamics (*Figure 2*), and happens when plus ends enter the nuclear membrane bridge formed after the dumbbell transition (*Figure 3*). Furthermore, microtubules do not switch to the slow growing state if bridge formation is prevented by cerulenin treatment or Aurora B inhibition (*Figure 4*). Our data suggests that microtubule growth speed is mainly governed by spatial cues, rather than a 'timer' (*Figure 2K*, *Figure 2—figure supplement 3B*), which is reminiscent of recent studies showing that late mitotic events respond to spindle length and not time (*Afonso et al., 2014*; *Afonso et al., 2019*).

It is tempting to speculate on mechanisms that could decrease microtubule growth speed when plus ends enter the nuclear membrane bridge. ase1 and les1 deletions reduce the decrease in growth speed associated with internalisation of microtubules in the nuclear membrane bridge (*Figure 5F*, *Figure 4—figure supplement 2*) while also perturbing the sorting of nuclear pore complexes to the center of the nuclear membrane bridge (*Expósito-Serrano et al., 2020*; *Dey et al., 2020*). It is therefore possible that proteins recruited to the center of the membrane bridge might directly reduce microtubule growth speed. Alternatively, close contact of microtubules with the nuclear membrane or its associated proteins might physically hinder growth (*Dogterom and Yurke, 1997*; *Tischer et al., 2009*). Finally, since there is no diffusion between the nuclear membrane bridge and the daughter nuclei (*Lucena et al., 2015*; *Boettcher et al., 2012*), the availability of tubulin dimers or another factor could limit microtubule growth when plus ends enter the bridge. Consistent with this last possibility, in *S. cerevisiae* the nuclear bridge acts as a diffusion barrier between daughter nuclei only in the presence of Ase1 (*Boettcher et al., 2012*). It would be interesting to test whether microtubule growth speed also decreases during anaphase in the closely related fission yeast *S. japonicus*, which does not form a nuclear membrane bridge during anaphase (*Yam et al., 2011*). Importantly, membranes also tightly wrap around the spindle in other systems, such as the midbody in animal cells (*Hu et al., 2012*), or the phragmoplast in plants (*Steiner et al., 2016*), and may affect microtubule dynamics. Nevertheless, at least in fission yeast, we could not here assign a biological function to this decrease in microtubule growth speed, as completely preventing it (*Figure 4*), delaying it (*Figure 3—figure supplement 1E*) or reducing its extent (*Figure 4—figure supplement 2*) did not compromise spindle stability nor overall microtubule organisation.

## Cross-talk between microtubule sliding and growth

We saw no evidence of cross-talk between microtubule sliding and growth. Klp9 has no noticeable effect on microtubule growth speed, despite its growth promoting activity in monopolar spindles (*Krüger et al., 2021*). We have not seen growth-limited sliding in unperturbed spindles either: although microtubule growth speed decreases by ~60% (*Figure 1F*), spindle elongation speed remains constant throughout anaphase B (*Figure 1—figure supplement 1A*). In *S. cerevisiae* and *Drosophila*, deletion or depletion of Kinesin-8, which decreases catastrophe rate (*Wang et al., 2010*; *Rizk et al., 2014*), increases sliding speed and final spindle length (*Wang et al., 2010*; *Rizk et al., 2014*). This could indicate that sliding is growth-limited in these organisms. On the other hand, in *S. cerevisiae*, deletion of kinesin-8 kip3 also delays spindle disassembly (*Woodruff et al., 2010*). Other mutations that delay spindle disassembly (like cdh1Δ, which prevents the degradation of midzone crosslinkers) produce hyperelongated spindles, similarly to kip3 deletion (*Woodruff et al., 2010*). In both kip3Δ and cdh1Δ, spindle elongation only stops when spindles are cut by the cytokinetic ring (*Woodruff et al., 2010*), suggesting that no mechanism exists to stop sliding during anaphase B other than the disassembly of the midzone, and the same has been observed in *S. pombe* (*Lucena et al., 2015*). Growth-limited sliding does not seem to occur in HeLa cells either, as depletion of kinesin-4 does not affect spindle elongation velocity (*Vukušić et al., 2021*). Ultimately, measuring microtubule dynamics in these systems will reveal their impact on microtubule sliding. Nevertheless, growth-limited sliding is widely conserved, and may contribute to spindle stability in *S. pombe* when microtubule dynamics are perturbed, such as during starvation (*Tanabe et al., 2020*).

## Promoting microtubule rescues at the midzone edges grants coordination of sliding and net polymerisation

The microtubule rescue factor Cls1 (CLASP) is recruited to the midzone by the crosslinker Ase1 (PRC1) in several species (*Liu et al., 2009*; *Bratman and Chang, 2007*; *Kitazawa et al., 2014*), and it had been previously proposed that this would restrict rescues to the midzone (*Bratman and Chang, 2007*). Indeed, we observed that rescues occur most frequently at midzone edges (*Figure 1G*). Thus, spindle microtubules remain stable from the pole to the midzone edge, and only their distal parts including the plus ends alternate between assembly and disassembly (*Figure 7A*). Simulations show that promoting microtubule rescue at midzone edges is sufficient to maintain the overlap between microtubules and sustain sliding across a wide range of microtubule growth speeds (*Figure 6F*), indicating that this mechanism is robust against perturbations on microtubule growth. Our study suggests that *S. pombe* cells adopted this mechanism to ensure spindle integrity, rather than a precise regulation of microtubule growth. This is supported by the fact that deletion of Ase1 (*Loiodice et al., 2005*; *Yamashita et al., 2005*) and inactivation of Cls1 (*Bratman and Chang, 2007*) lead to spindle collapse, while perturbing the normal microtubule growth speed evolution during anaphase B has no effect on spindle stability or organisation (*Figure 4*). In animal cells, the situation is likely more complex: in HeLa cells, depletion of kinesin-4 leads to highly dynamic interpolar microtubules (*Hu et al., 2011*), but spindles still elongate with normal speed during anaphase B (*Vukušić et al., 2021*), indicating that a precise regulation of microtubule growth is not required. However, depletion of PRC1/Ase1 does not lead to spindle collapse during anaphase B either (*Vukušić et al., 2021*; *Pamula et al., 2019*), suggesting that if microtubule rescue organisation is required for spindle stability, it may rely on a mechanism other than recruitment of CLASP by PRC1. It would be important to understand the relative contribution to spindle stability of these mechanisms, and others not present in yeast during anaphase B, like microtubule nucleation (*Uehara et al., 2009*; *Uehara and Goshima, 2010*).

We do not know the molecular mechanism by which rescues happen more often at midzone edges. We can speculate that this might be related to the fact that protofilaments curl outwards when microtubules shrink (*McIntosh et al., 2008*). Indeed, the distances between adjacent microtubules in the midzone are comparable to the radius of curling protofilaments (*McIntosh et al., 2008*; *Ward et al., 2014*), such that steric collisions alone could promote protofilament straightening, enhancing rescue similarly to what has been proposed for kinetochore protein Dam1 (*Franck et al., 2007*; *McIntosh et al., 2008*). Recent in vitro studies have combined PRC1, CLASP and dynamic microtubules (*Hannabuss et al., 2019*; *Mani et al., 2021*), so it would be interesting to see whether in a reconstitution resembling the yeast spindle rescues also happen at the edges, which would indicate that this is an inherent property of PRC1/CLASP systems.

In summary, a mechanism to coordinate microtubule sliding and growth relying on rescue localisation is robust against perturbations in polymerisation speed and allows for an array of antiparallel microtubules with a given overlap to continuously elongate. Given the diversity of sizes and morphologies of spindles, and the fact that growth limited sliding is widely conserved, other mechanisms are likely to be discovered in the future.

# Materials and methods

## Key resources table

| Reagent type (species) or resource | Designation | Source or reference | Identifiers | Additional information |
|---|---|---|---|---|
| Gene (*S. pombe*) | ade6 | Pombase | SPCC1322.13 | |
| Gene (*S. pombe*) | alp7 | Pombase | SPAC890.02c | |
| Gene (*S. pombe*) | ark1 | Pombase | SPCC320.13c | |
| Gene (*S. pombe*) | ase1 | Pombase | SPAPB1A10.09 | |
| Gene (*S. pombe*) | atb2 | Pombase | SPBC800.05c | |
| Gene (*S. pombe*) | cdc25 | Pombase | SPAC24H6.05 | |
| Gene (*S. pombe*) | clp1 | Pombase | SPAC1782.09c | |

*Continued on next page*

*Continued*

| Reagent type (species) or resource | Designation | Source or reference | Identifiers | Additional information |
|---|---|---|---|---|
| Gene (*S. pombe*) | cls1 | Pombase | SPAC3G9.12 | |
| Gene (*S. pombe*) | cut11 | Pombase | SPAC1786.03 | |
| Gene (*S. pombe*) | dis1 | Pombase | SPCC736.14 | |
| Gene (*S. pombe*) | imp1 | Pombase | SPBC1604.08c | |
| Gene (*S. pombe*) | klp5 | Pombase | SPBC2F12.13 | |
| Gene (*S. pombe*) | klp6 | Pombase | SPBC1685.15c | |
| Gene (*S. pombe*) | klp9 | Pombase | SPBC15D4.01c | |
| Gene (*S. pombe*) | les1 | Pombase | SPAC23C4.05c | |
| Gene (*S. pombe*) | leu1 | Pombase | SPBC1A4.02c | |
| Gene (*S. pombe*) | mal3 | Pombase | SPAC18G6.15 | |
| Gene (*S. pombe*) | nem1 | Pombase | SPBC3B8.10c | |
| Gene (*S. pombe*) | sid4 | Pombase | SPBC244.01c | |
| Gene (*S. pombe*) | ura4 | Pombase | SPCC330.05c | |
| Strain, strain background (*S. pombe* 972) | wt | Lab collection | TP5567 | h- alp7-3xGFP:kanMX6 leu1:sid4-GFP leu1-32 ura4-D18 |
| Strain, strain background (*S. pombe* 972) | wt | Lab collection | TP3288 | h- cls1-3xGFP:kanMX6 mCherry-atb2:hphMX6 leu1-32 ura4-D18 |
| Strain, strain background (*S. pombe* 972) | wt | Lab collection | TP5986 | h + kanMX6:P1nmt1-GFP-mal3 leu1:sid4-GFP |
| Strain, strain background (*S. pombe* 972) | mal3Δ | Lab collection | TP5624 | h- alp7-3xGFP:kanMX6 leu1:sid4-GFP mal3Δ:hphMX6 leu1-32 ura4-D18 |
| Strain, strain background (*S. pombe* 972) | klp9Δ | Lab collection | TP5574 | h + alp7-3xGFP:kanMX6 leu1:sid4-GFP klp9Δ:natMX6 leu1-32 ura4-D18 |
| Strain, strain background (*S. pombe* 972) | clp1Δ | Lab collection | TP5668 | h- alp7-3xGFP:kanMX6 leu1:sid4-GFP clp1Δ:natMX6 leu1-32 ura4-D18 |
| Strain, strain background (*S. pombe* 972) | klp5Δklp6Δ | Lab collection | TP5672 | h + alp7-3xGFP:kanMX6 leu1:sid4-GFP klp5Δ:Ura Klp6Δ:Hph leu1-32 ura4-D18 |
| Strain, strain background (*S. pombe* 972) | dis1Δ | Lab collection | TP5670 | h + alp7-3xGFP:kanMX6 leu1:sid4-GFP Dis1Δ:hphMX6 leu1-32 ura4-D18 |
| Strain, strain background (*S. pombe* 972) | cls1OE | Lab collection | TP5625 | h- alp7-3xGFP:kanMX6 leu1:sid4-GFP natMX6:P1nmt1-cls1 leu1-32 ura4-D18 |
| Strain, strain background (*S. pombe* 972) | cls1off | Lab collection | TP5572 | h + alp7-3xGFP:kanMX6 leu1:sid4-GFP natMX6:P81nmt1-cls1 leu1-32 ura4-D18 |
| Strain, strain background (*S. pombe* 972) | cls1-3xGFPOE | Lab collection | TP5499 | h- natMX6:P1nmt1-cls1-3xGFP:kanMX6 mcherry-atb2:hphMX6 leu1-32 ura4-D18 ade6- |
| Strain, strain background (*S. pombe* 972) | cls1-3xGFPoff | Lab collection | TP5337 | h- natMX6:P81nmt1-cls1-3xGFP:kanMX6 mCherry-atb2:hphMX6 leu1-32 ura4-D18 ade6- |
| Strain, strain background (*S. pombe* 972) | wt/ ctrl/ cerulenin | Lab collection | TP5662 | h + alp7-3xGFP:kanMX6 leu1:sid4-GFP cut11-mCherry:hphMX6 leu1-32 |
| Strain, strain background (*S. pombe* 972) | cdc25-22 | Lab collection | TP5663 | h alp7-3xGFP:kanMX6 leu1:sid4-GFP cut11-mCherry:hph cdc25-22 leu1-32 ura4-D18 |
| Strain, strain background (*S. pombe* 972) | klp9OE | Lab collection | TP5665 | h + alp7-3xGFP:kanMX6 leu1:sid4-GFP cut11-mCherry:hph natMX6:P1nmt1-klp9 leu1-32 ura4-D18 |
| Strain, strain background (*S. pombe* 972) | wt | Lab collection | TP5717 | h + alp7-3xGFP:kanMX6 leu1:sid4-GFP cut11-mCherry:natMX6 leu1-32 ura4-D18 |
| Strain, strain background (*S. pombe* 972) | ark1-as3 | Lab collection | TP5761 | h + alp7-3xGFP:kanMX6 leu1:sid4-GFP cut11-mCherry:natMX6 ark1-as3:hphMX6 leu1-32 ura4-D18 |

*Continued on next page*

*Continued*

| Reagent type (species) or resource | Designation | Source or reference | Identifiers | Additional information |
|---|---|---|---|---|
| Strain, strain background (*S. pombe* 972) | ase1Δ | Lab collection | TP5577 | h- alp7-3xGFP:kanMX6 leu1:sid4-GFP ase1Δ:natMX6 leu1-32 ura4-D18 |
| Strain, strain background (*S. pombe* 972) | ase1off | Lab collection | TP5570 | h + alp7-3xGFP:kanMX6 leu1:sid4-GFP natMX6:P81nmt1-Ase1 leu1-32 ura4-D18 |
| Strain, strain background (*S. pombe* 972) | mcherry-ase1OE | Lab collection | TP5772 | h- alp7-3xGFP:natMX6 leu1:sid4-GFP kanMX6:P1nmt1-mCherry-ase1 leu1-32 ura4-D18 |
| Strain, strain background (*S. pombe* 972) | wt | Lab collection | TP5842 | h- alp7-3xGFP:natMX6 leu1:sid4-GFP mCherry-ase1::kanMX6 leu1-32 ura4-D18 |
| Strain, strain background (*S. pombe* 972) | ase1Δ | Lab collection | TP5836 | h- alp7-3xGFP:kanMX6 leu1:sid4-GFP cut11-mCherry:hph ase1Δ:natMX6 leu1-32 ura-D18 |
| Strain, strain background (*S. pombe* 972) | imp1Δ | Lab collection | TP5981 | h + alp7-3xGFP:kanMX6 leu1:sid4-GFP cut11-mCherry:hph imp1Δ:ura4 leu1-32 ura-D18 |
| Strain, strain background (*S. pombe* 972) | les1Δ | Lab collection | TP5982 | h + alp7-3xGFP:kanMX6 leu1:sid4-GFP cut11-mCherry:natR les1Δ:Hph leu1-32 ura4-D18 |
| Strain, strain background (*S. pombe* 972) | nem1Δ | Lab collection | TP5723 | h + alp7-3xGFP:kanMX6 leu1:sid4-GFP cut11-mCherry:hph nem1Δ:NatMx6 leu1-32 |
| Recombinant DNA reagent | pFA6a-natMX6-Pnmt81 (plasmid) | Lab collection | pSR176 | see *Supplementary file 1* |
| Recombinant DNA reagent | pFA6a-natMX6-P1nmt1 (plasmid) | Lab collection | pSR174 | see *Supplementary file 1* |
| Recombinant DNA reagent | pFa6a-kanMX6-P1nmt1-mCherry (plasmid) | Ken Sawin's lab | pKS394 | |
| Recombinant DNA reagent | pFa6a-kanMX6-P41nmt1-mCherry (plasmid) | Ken Sawin's lab | pKS395 | |
| Recombinant DNA reagent | pFa6a-mCherry-ase1:kanMX6 (plasmid) | Lab collection | pML1 | see *Supplementary file 1* |
| Sequence-based reagent | SR6.78 | Eurofins France | SR6.78 | GCGAGTTTTT GCGAG TTTTT AATAT TCTCT TCGCA AACAA CGCTT CACGT TTCTC TTGTT TCGCT CGTTT CATCA ATATA TTTGT AATTG GAATT CGAGC TCGTT TAAAC |
| Sequence-based reagent | SR6.79 | Eurofins France | SR6.79 | TCAGTATATA TCAGT ATATA GATGA AAGCT TTAGA ATTTC ATACC ATTAC TTTTA AGGAA CTTTA AAAAA TCTTG CGCAT CCTTA TCCGC CATGA TTTAA CAAAG CGACT ATA |
| Sequence-based reagent | SR5.56 | Eurofins France | SR5.56 | CTTTTATGAA CTTTT ATGAA TTATC TATAT GCTGT ATTCA TATGC AAAAA TATGT ATATT TAAAT TTGAT CGATT AGGTA AATAA GAAGC GAATT CGAGC TCGTT TAAAC |
| Sequence-based reagent | SR5.57 | Eurofins France | SR5.57 | AGTTTTCATA AGTTT TCATA TCTTC CTTTA TATTC TATTA ATTGA ATTTC AAACA TCGTT TTATT GAGCT CATTT ACATC AACCG GTTCA GAATT CGAGC TCGTT TAAAC |
| Sequence-based reagent | ML31 | Eurofins France | ML31 | CGTTGTATAC CGTTG TATAC TTTGT ATGCA TCGCT TCTTT TGGTG AATTT TTTAA TTCTT TGCAA TCGCA GCAGA GAGAA AATAA TTGTA CGGAT CCCCG GGTTA ATTAA |
| Sequence-based reagent | ML32 | Eurofins France | ML32 | AAGTTATTTT AAGTT ATTTT AGACC ATCGT TACTG GTGAT AAATA ACGAG TAAAT TACTC ACGAA AAAAA AAAGG AATCA TGAAA AGCAC GAATT CGAGC TCGTT TAAAC |
| Sequence-based reagent | ML39 | Eurofins France | ML39 | TTAGA TTCAT TATTA GAGTG ATTAT CTTTT TCAGC AATAG AATCA GTGCT TTGAA TGTCA TCCAT CATTA CTGTT TGCAT CTTGT ACAGC TCGTC CATGC |
| Sequence-based reagent | ML40 | Eurofins France | ML40 | TAAGC AGTCG ACATG GTGAG CAAGG GCGAG |
| Sequence-based reagent | ML41 | Eurofins France | ML41 | TAAGC AGGCG CGCCT TAAAA GCCTT CTTCT CCCCA TTCA |

*Continued on next page*

*Continued*

| Reagent type (species) or resource | Designation | Source or reference | Identifiers | Additional information |
|---|---|---|---|---|
| Sequence-based reagent | ML42 | Eurofins France | ML42 | AGTTT TCATA TCTTC CTTTA TATTC TATTA ATTGA ATTTC AAACA TCGTT TTATT GAGCT CATTT ACATC AACCG GTTCA ATGGT GAGCA AGGGC GAG |
| Chemical compound, drug | cerulenin | Sigma-Aldrich | C2389 | |
| Chemical compound, drug | 1NM-PP1 | Sigma-Aldrich | 529,581 | |
| Software, algorithm | MATLAB | Mathworks | | |
| Software, algorithm | Univarscatter | GitHub | | https://github.com/manulera/UnivarScatter |
| Software, algorithm | hline-vline | MathWorks | | https://www.mathworks.com/MATLABcentral/fileexchange/1039-hline-and-vline |
| Software, algorithm | geom2d | GitHub | | https://www.mathworks.com/MATLABcentral/fileexchange/7844-geom2d |
| Software, algorithm | simulation | GitHub | https://github.com/manulera/simulationsLeraRamirez2021 | |
| Software, algorithm | Imagej | Imagej | https://imagej.net/software/fiji/ | |
| Software, algorithm | KymoAnalyzer | GitHub | https://github.com/manulera/KymoAnalyzer | |

## Production of *S. pombe* Mutant Strains

All used strains are isogenic to wild-type 972 and were obtained from genetic crosses, selected by random spore germination and replicated on plates with corresponding drugs or supplements. Gene deletion and tagging was performed as described previously (*Bähler et al., 1998*), following *Kawai et al., 2010*. All strains, oligonucleotides and plasmids used, and how they were made are described in the *Supplementary file 1*.

## Fission yeast culture

All *S. pombe* strains were maintained at 25 °C in YE5S plates and refreshed every third day. One day before the microscopy experiments, cells were transferred to liquid YE5S culture, and imaged the next day at exponential growth. For all experiments except for *Figure 3*, the cells were grown overnight in YE5S liquid medium at 25 °C. For experiments in the absence of thiamine (used to induce overexpression of klp9, *Figure 3*), cells were pre-grown in liquid YE5S, then washed three times with deionised water, transferred to EMM supplemented with adenine, leucine and uracil, and incubated 18–22 hr at 25 °C prior to the microscopy experiment. For cerulenin treatment, cells were incubated for 1 hr in liquid medium with the indicated cerulenin concentration prior to the experiment (Sigma-Aldrich, stock solution was 10 mM in DMSO), the same volume of DMSO was added in control. For Aurora B inactivation, 5 µM of 1NM-PP1 was added to both wild-type and ark1-as3 cells (Sigma-Aldrich, stock solution was 5 mM in DMSO), and cells were imaged immediately.

## Live-cell microscopy

For live-cell imaging, cells were mounted on YE5S agarose pads, containing 4% agarose (*Tran et al., 2004*). For the experiments in *Figure 4*, the drug or DMSO was added at the same concentration in the agarose pad as in the liquid medium. Movies used for kymographs were acquired with Structural Illumination Microscopy (SIM).

Imaging of data from figures *Figure 1*, *Figure 1—figure supplement 1*, *Figure 2* (mal3, cls1, klp9 and their corresponding wt) and *Figure 5—figure supplement 3* (ase1off and its corresponding wt) was performed at 27 °C with a Nikon Eclipse Ti inverted microscope, equipped with a Nikon CFI Plan Apochromat 100x/1.4 NA objective lens, a Nikon Perfect Focus System (PFS), a Mad City Labs integrated Nano-View XYZ micro- and nano-positioner, a Gataca Systems LIVE-SR SIM module, a Yokogawa Spinning Disk CSU-X1 unit, a Photometrics Prime 95B sCMOS camera, controlled by MolecularDevices MetaMorph 8.0. For GFP and mCherry imaging, solid-state lasers of 488 nm (100 mW) and 561 nm (50 mW) were used.

The rest of the imaging was performed at 27 °C with a Nikon Eclipse Ti2 inverted microscope, equipped with a Nikon CFI Plan Apochromat 100x/1.4 NA objective lens, a Nikon Perfect Focus System (PFS), a Mad City Labs integrated Nano-View XYZ micro- and nano-positioner, a Gataca Systems LIVE-SR SIM module, a Yokogawa Spinning Disk CSU-W1 unit, a Photometrics Prime 95B sCMOS camera, controlled by MolecularDevices MetaMorph 8.0. For GFP and mCherry imaging, solid-state lasers of 488 nm (150 mW) and 561 nm (150 mW) were used.

Movies for kymographs were acquired as follows: in the GFP channel, images were acquired as stacks of 5 planes spaced 0.5 µm without binning every 4 s during 15 min (except for mCherry-Ase1 movies, where the images were acquired every 5 s). Exposure was 100ms (Gain 3). In the mCherry channel, for Cut11-mCherry images were acquired as single stacks, every 16 s without binning during 15 min. For mCherry-ase1, images were acquired as stacks of 5 planes spaced 0.5 µm without binning every 8 s during 15 min. Exposure was 200ms (Gain 3). Movies where intensity measurements were made were acquired as stacks of 11 planes spaced 0.5 µm without binning every minute during 90 min. Exposure was 100ms (Gain 3) in all channels. The remaining movies were acquired as stacks of 7 planes spaced 1 µm without binning every minute during 90 min. Exposure was 100ms (Gain 3) in all channels.

To minimise the inter-experiment variance, all the data shown within the same plot was acquired in parallel. In the case of movies used for kymographs this meant that samples of the conditions tested were alternated on the microscope. In the case of one minute interval movies, positions corresponding to all conditions were imaged simultaneously.

## Statistical analysis

All statistical analysis was performed using MATLAB. In the main text, values are reported as mean ± standard deviation. To avoid pseudo-replication and use the right degrees of freedom to calculate statistical significance we used linear mixed models (see *Lazic et al., 2018*; *Aarts et al., 2015* for detailed explanation), and degrees of freedom were estimated with Satterthwaite approximation. In essence, linear mixed models include random effects to account for the fact that multiple measurements from the same experiment or cell are not independent. This is important for our nested data where several measurements of a given magnitude (e.g. microtubule growth) are taken from various cells in several experiments. Additionally, in some cases these measurements are classified in categories (before, outside, inside).

To study the fixed effect of a 'condition' (genetic background or drug treatment) on a given response variable $y$ for binned data (such as *Figure 2G–L*) or univariate distributions (*Figure 1— figure supplement 2C, F, G*), we took into account the random effects associated with cell and experiment variability. The data was fitted to the following mixed-effects model (in MATLAB / R notation): `y ~ condition + (condition|experiment) + (1|cell:experiment)` In this model, `(condition|experiment)` accounts for the fact that the effect of 'condition' may vary between experiments, and `(1|cell:experiment)` accounts for cell variability, nested within the experiment. The term `(1|cell:experiment)` is not used in cases where there is only one measurement per cell (*Figure 5D*, *Figure 3—figure supplement 1A, C, D*) or when comparing interphase and anaphase cells (*Figure 1—figure supplement 2F, G*).

To study the fixed effect of 'condition' (genetic background or drug treatment) and 'position' (before / outside / inside, see cartoons in *Figure 4*) on microtubule growth speed, we took into account the random effects associated with cell and experiment variability. The data was fitted to the following mixed-effects model (in MATLAB / R notation): `growthspeed ~ condition * position + (condition|experiment) + (position|cell:experiment)` In this model, (condition|experiment) accounts for the fact that the effect of 'condition' may vary between experiments, and (position|cell:experiment) accounts for the fact that the effect of 'position' may vary between cells (which are nested within an experiment). These mixed effects models were used to generate the summary statistical tables in the supplementary. For calculating the mean and confidence intervals of the mean, we used MATLAB's predict function. To calculate the percentage of variability that is explained by the categorisation 'before', 'inside' and 'outside' in *Figure 3* (mentioned in main text) we fitted the data to a conventional linear model `(speed ~ position)`, and reported the $R^2$ as the percentage of variability explained by the categorisation. To compare microtubule growth speed 'outside' and 'inside' in 'ark1-as3 abnormal' cells (see main text) we reported a p-value obtained with

MATLAB's coefTest function, which evaluates the significance of the difference between means for any pair of categories in the model. We provide our MATLAB analysis script in the supplementary files (Source Code File 1, statistical_analysis.m) to reproduce this statistical analysis. The values of standard deviation reported in the main text and in *Table 1* are the average of the standard deviations of individual experiments.

## Image and data analysis

All feature detections, kymographs and length measurements were done in maximal projections of fluorescence microscopy images. Intensity measurements were performed on summed projections.

### Spindle detection

Sid4-GFP Alp7-3xGFP and fluorescence tubulin movies were pre-processed with the Fiji distribution of Imagej (https://imagej.net/Fiji). The plugin *Trainable Weka Segmentation* (*Arganda-Carreras et al., 2017*) was used to generate probability images of each frame in maximal projections. Each probability image has the same size as the original maximal projection, with values in [0,1]. Higher values correspond to higher likelihood of that pixel corresponding to a spindle. The probability images were later used with a custom MATLAB script to detect spindles, which is publicly available (link). The user is required to draw the profile of each analysed cell by hand. Since generally cells do not move between frames, the same profile can be used for all the frames, but it can be changed between frames if necessary.

To find the spindles, we use the pixels inside the cell mask in probability images for which the probability is bigger than 0.8 (This gives a good segmentation of the spindle). Pixels are defined by position $(X_i, Y_i)$ and probability $(P_i)$. We find the parameters $(X_0, Y_0, \theta, \alpha)$ that maximise a functional $F$ defined as:

$$\begin{bmatrix} x_i \\ y_i \end{bmatrix} = \begin{bmatrix} cos\theta & -sin\theta \\ sin\theta & cos\theta \end{bmatrix} \begin{bmatrix} X_i - X_0 \\ Y_i - Y_0 \end{bmatrix} \tag{1}$$

$$F(x_i, y_i, P_i) = \sum_{i=i}^{N} P_i^2 \, exp\left(-\left|\frac{\alpha x_i^2 - y_i}{\lambda}\right|\right) \tag{2}$$

*Equation 1* rotates a point by angle $\theta$ around $(X_0, Y_0)$. *Equation 2* defines an objective function, aiming to fit a spindle defined by $g(x) = \alpha x^2$, a polynomial of degree 2 in the rotated axis. F is a weighted sum computed from the probability $P_i$ and exponential terms derived from the distance to $g(x)$, with a characteristic width $\lambda$=0.55 μm. To find the polynomial that best matches the spindle, we find the values of $(X_0, Y_0, \theta, \alpha)$ that maximise $F$. Then, to find the spindle edges, we project all the points of the probability image on the polynomial $g(x)$. The edges are defined as the two points on $g(x)$ that contain all projections with $P_i > 0.6$ between them. We define the length of the spindle as the length of the curve $g(x)$ between these two edges. We fit data in each frame to this function, starting by the first time point (2.1). Initially, we set $\alpha$ to zero, until we find a spindle of length greater than 6 μm, since shorter spindles are straight and well described by a line. If the algorithm fails to find the trace of the spindle, the user can draw the spindle by hand. In Alp7-3xGFP Sid4-GFP movies used for kymographs, we use a slight variation in which first the SPBs are detected, since Sid4-GFP signal is stronger, and we constrain the fit to polynomials passing by those two points.

### Intensity measurements

To measure the intensity of fluorescent tubulin and MAPs, we used intensity profiles (MATLAB *improfile* function) on summed projections along curves obtained with the spindle detection algorithm described above. The curve points were evenly spaced at a distance of 1 pixel. We used seven parallel curves, separated by 1 pixel, the central one coinciding with the orthogonal polynomial $g(x)$. The total width corresponds to 0.77 μm in our microscope. We consider this to be the signal region. As background, we subtracted the median of two parallel regions of 4 pixels (0.44 μm) width flanking the signal region. The intensity was the sum of the intensities of all the points in the signal region after background subtraction. For the density of Cls1-3xGFP (*Figure 2—figure supplement 2B, D*), we only measured intensities of Cls1-3xGFP and mCherry-Atb2 in the central 2 μm of the spindle.

## Kymograph analysis

To generate kymographs, we used intensity profiles (MATLAB *improfile* function) on maximal projections of Sid4-GFP Alp7-3xGFP movies on which a Gaussian filter of 1 pixel width was applied, along curves obtained with the spindle detection algorithm described above. For each of the frames in the movie, we constructed a maximal projection along a curve calculated from the fit, where points were evenly spaced at a distance of 1 pixel. We projected all the points that were at a distance of 3 or less pixels from the spindle trace, and this yielded a 1-D array of intensities for each time point. To align the arrays into a kymograph we used the center of mass of the maximal projection of the movie in time as a reference, and we aligned all the arrays such that this point would be at the center of the kymograph. Cut11-mCherry and mCherry-Ase1 kymographs were constructed the same way, but no Gaussian filter was applied. In the figures, kymographs were realigned to have the spindle center at the center of the kymograph.

To annotate the kymographs, we developed a MATLAB program, KymoAnalyzer (https://github.com/manulera/KymoAnalyzer), allowing one to draw polygonal lines to mark the poles of the spindle, the microtubule growth or shrinking events in the Sid4-GFP Alp7-3xGFP channel, and the membrane bridge edges in the Cut11-mCherry channel. The manually drawn lines are then resampled by interpolating the position at every time point. Microtubule shrinking events had a lower Alp7-3xGFP signal, so they were not always detectable after a growth event. However, they were always preceded by a growth event with higher intensity (see *Figure 1C*), which allowed us to discriminate between growth and shrinkage events. In order to assign a growing microtubule plus end to a spindle pole, we calculated the growth speed with respect to both spindle poles (see below), and assigned the plus end to the spindle pole that gave the highest speed. Shrinking plus ends were assigned to a pole based on the growth event that preceded them. Only events that were clearly resolvable were considered. The ones where superimposition of comets prevented resolving them were not included. Duration of the growth event was defined as the time between apparition of the comet and its disappearance. To calculate the microtubule growth speed, the length vs. time curve was fitted to a first order polynomial using MATLAB *polyfit*. To calculate the spindle elongation speed (*Figure 3—figure supplement 1C*) the spindle length (distance between SPBs) in time was fitted to a first order polynomial using MATLAB *polyfit*. The spindle center is defined as the middle between the poles. Kymographs of Cut11-mCherry were also used to determine the spindle length at which dumbbell transition occurred (*Figure 3—figure supplement 1D*).

## Anaphase onset determination

To determine the onset of anaphase, we fitted the curves of length of spindle in time to the function $G(t)$, with fitting parameters ($L_0$, $t_1$, $t_2$, $s_1$, $s_2$, $s_3$):

$$G(t) = \begin{cases} L_0 + t\, s_1 & \text{if } t \leq t_1 \\ G(t_1) + (t - t_1)\, s_2 & \text{if } t > t_1 \text{ and } t \leq t_2 \\ G(t_2) + (t - t_2)\, s_3 & \text{if } t > t_2 \end{cases} \tag{3}$$

$G(t)$ is a series of three linear fits that start at $L_0$ at $t = 0$. Anaphase onset is determined by $t_2$. In general, fits correctly identified the switch in velocity from Phase II to III seen in *Figure 1—figure supplement 1A*. In some cases, particularly in klp9Δ, where the anaphase velocity decreases in time, $t_2$ was fixed manually if the fit gave an obviously wrong result.

## Inference of anaphase time from spindle length

In *Figure 2—figure supplement 3B* and *Figure 3—figure supplement 1E* we infer anaphase timing from spindle length. To do this, we use the curves of spindle length in time obtained in movies in which images were acquired every minute. We fit the curves of spindle length in time during anaphase to first order polynomials using MATLAB *polyfit* (*Figure 2—figure supplement 3C*, *Figure 3—figure supplement 1H*), with anaphase onset determined as indicated above. Using the fit parameters, we then transform the spindle length measured in kymographs into time.

## Public MATLAB libraries used in this study

- geom2d - 2D geometry library

- hline-vline - plotting tool
- UnivarScatter - univariate scatter plots

## Computational model

The simulation is written in Python and available under an open source license https://github.com/manulera/simulationsLeraRamirez2021.

We consider a one-dimensional model in which all microtubules are straight and aligned with the X axis, with the origin corresponding to the spindle center. The spindle poles, which contain the microtubule minus ends, are positioned symmetrically around the origin. Microtubules growing from one pole are orientated towards the opposite pole (*Figure 6A*). Their positions in the YZ plane are recorded by an index representing their position on a transverse chequerboard lattice (*Figure 6—figure supplement 1A*). This index may change when the spindle 'reorganises' (*Figure 6—figure supplement 1A*, and see below). For simplicity, we do not simulate crosslinkers or other MAPs, and we assume that the midzone is a region of the space centered at the origin that has a constant length ($L_m$). In each simulation, $L_m$ is drawn randomly, sampling from a fit of the midzone length data shown in *Figure 1G* to a normal distribution (*Figure 6—figure supplement 1C*). Each microtubule has its own length, and can exist in a growing or shrinking state. Microtubule growth, shrinkage and sliding occur at constant speeds ($v_p$, $v_d$, $v_s$ *Table 2*). The duration of microtubule growth events is sampled from a fit of the cumulative distribution of duration of microtubule growth events shown in *Figure 1—figure supplement 1C* to:

$$(1 - e^{-\theta t})^n \tag{4}$$

This function represents the probability of $n$ events occurring in an interval of time $t$, if they are independent and occur with a constant rate $\theta$. For $\theta$=3.17 min$^{-1}$ and $n$=8.53 (see *Table 2*), we obtained a good agreement with the data (*Figure 6—figure supplement 1D*). This suggests that catastrophe is not a single step process, agreeing with what was proposed in *Gardner et al., 2011*. We assume for simplicity that a growing microtubule reaching the spindle pole immediately undergoes catastrophe. Shrinking microtubules may be rescued if their plus end is inside the midzone. Based on the fact that Cls1 is recruited by Ase1 (*Bratman and Chang, 2007*) and that Cls1 levels remain constant during anaphase (*Figure 2—figure supplement 2A*), we assume that a fixed amount of rescue factor is distributed along the overlap between microtubules. Therefore, the integral of rescue rate ($r$) along the overlap length of the midzone is constant, and given by $R = N \int_{l=0}^{l=L_m} r \, dl$, where N is the number of microtubule pairs in the square lattice. The rescue rate of a microtubule with $n$ neighbours is given by the following expression:

$$r = \frac{nR}{L_m N} P(x, \alpha, \beta) \tag{5}$$

$P(x, \alpha, \beta)$ is the probability density of a beta distribution with parameters $\alpha$ and $\beta$ as a function of $x$, a reduced coordinate that maps the position of microtubule plus ends in the midzone from zero to one, with zero corresponding to the midzone edge closest to the spindle pole (*Figure 6A*). The beta distribution offers differently shaped distributions of rescue activity while keeping $R$ constant. The spindle can reorganise when a microtubule is lost by complete depolymerisation. We assume that another microtubule can rearrange in the YZ plane to maximise antiparallel connections, as these are the arrangements observed by electron microscopy (*Ward et al., 2014*; *Ding et al., 1993*). Microtubules move to another position of the square lattice, of the same orientation, to increase the compactness of the structure. This can happen in two cases, as shown in *Figure 1—figure supplement 1A*. (1) A microtubule is lost and another one moves in its place (2) A neighbour of the lost microtubule moves to another available position. The moves only occur if they increase the number of antiparallel connections.

Simulations are initialised with a 4 µm long spindle, with 9 microtubules of 3 µm of length arranged in a square lattice as shown in *Figure 6A*. Initially they all start in a growing state. The system is evolved using a time step $h$ of 0.01 min. The new positions of spindle poles are calculated as $x_{t+h} = x_t + v_s h$, and microtubule lengths are updated similarly. Catastrophe and rescue are stochastic events simulated using the kinetic Monte Carlo method. At the beginning of the simulation, and whenever a microtubule rescue occurs, the time until the next catastrophe for a given microtubule ($\tau_c$) is sampled

from the probability distribution in *Equation 4*. At every time point, $h$ is subtracted from the $\tau_c$ of each growing microtubule, until $\tau_c$ is smaller than zero, at which point the microtubule switches to shrinkage. Shrinking microtubules can only be rescued inside the midzone, and at every time point the rescue rate ($r$) is given by *Equation 5*. The probability to be rescued during $h$, $1 - e^{-rh}$ is tested against a random number, uniformly distributed in [0,1]. At every time point, we check if the spindle 'collapses' by testing whether the longest microtubules of each pole are too short to overlap. The simulation finishes if the spindle breaks or when it reaches 20 min of simulated time.

The simulations of ase1 spindles are performed similarly, but the rescue rate is the same anywhere on the spindle and is given by $r = R/2L_t$, where $L_t$ is the total length of microtubules. In this model, we assume that a fixed amount of rescue factor is distributed all along microtubules and not along the overlap between neighbours. The factor 2 in the denominator ensures consistency between both models. The fitting parameters for *Equation 4* in these simulations are $\theta$=6.8 min$^{-1}$ and $n$=2.5 (*Table 2*).

## Comparison of simulations and experiments

To compare microtubule polymer length as a function of spindle length in simulations with total tubulin intensity as a function of spindle length in experiments (*Figure 6—figure supplement 1D*), we binned the data based on spindle length, in bins of 1 μm width. For each bin, we calculated the average value of polymerised tubulin in simulations ($s$) and tubulin intensity in experiments ($e$). Then we calculated a scaling factor ($f$) that minimised the difference between them, and used the sum of square differences across all $N$ bins ($\sum_{i=1}^{i=N}(s_i - e_if)^2$) to score the similarity between simulations and experiments.

To compare the distribution of spindle length at collapse in ase1$\Delta$ simulations and experiments (*Figure 6—figure supplement 1F*), we calculated their empirical cumulative probability distributions with MATLAB function *ecdf*. We then used the Kolmogorov-Smirnov distance between these distributions to score the similarity between simulations and experiments. The Kolmogorov-Smirnov distance is the supremum of the distances between cumulative probability distributions ($F$) across all their domain ($sup_x |F_1(x) - F_2(x)|$).

# Acknowledgements

We thank Lara Katharina Krüger and Serge Dmitrieff for discussions and critical reading of the manuscript. We thank Chuanhai Fu (University of Science and Technology of China), Stefania Castagnetti (LBDV, Sorbonne Université), Jonathan Millar (University of Warwick), Ken Sawin (University of Edinburgh), Sylvie Tournier (Université de Toulouse), Rafael R Daga (Universidad Pablo de Olavide, Sevilla) and Gautam Dey (EMBL Heidelberg) for kindly providing strains and plasmids used in this study. We thank Vincent Fraisier and Lucie Sengmanivong for the maintenance of microscopes at the PICT-IBiSA Imaging facility (Institut Curie), a member of the France-BioImaging national research infrastructure. We thank the Japan National BioResource Project – Yeast Genetic Resource Center (Osaka City University, Osaka University and Hiroshima University) for providing strains. Funding Manuel Lera-Ramirez was supported by a PhD fellowship from the European Union ITN-Divide Network, Marie Sklodowska-Curie Actions (grant number: 675737 s). This work and François Nédélec were supported by the Gatsby Charitable Foundation https://www.gatsby.org.uk/. This work is supported by grants from INCa, Fondation ARC, and La Ligue National Contre le Cancer - Ile de France. The Tran lab is a member of the Labex CelTisPhyBio, part of IdEx PSL.

# Additional information

## Funding

| Funder | Grant reference number | Author |
|---|---|---|
| H2020 Marie Skłodowska-Curie Actions | 675737 | Manuel Lera-Ramirez |
| Gatsby Charitable Foundation | | François J Nédélec |
| Institut National Du Cancer | | Phong T Tran |

| Funder | Grant reference number | Author |
|---|---|---|
| Fondation ARC pour la Recherche sur le Cancer | | Phong T Tran |
| La Ligue Contre le Cancer - Ile de France | | Phong T Tran |

The funders had no role in study design, data collection and interpretation, or the decision to submit the work for publication.

## Author contributions

Manuel Lera-Ramirez, Conceptualization, Data curation, Formal analysis, Investigation, Methodology, Software, Validation, Visualization, Writing - original draft, Writing – review and editing; François J Nédélec, Conceptualization, Software, Supervision, Writing – review and editing; Phong T Tran, Conceptualization, Funding acquisition, Project administration, Supervision, Writing – review and editing

## Author ORCIDs

Manuel Lera-Ramirez http://orcid.org/0000-0002-8666-9746
Phong T Tran http://orcid.org/0000-0002-2410-2277

## Decision letter and Author response

Decision letter https://doi.org/10.7554/eLife.72630.sa1
Author response https://doi.org/10.7554/eLife.72630.sa2

# Additional files

## Supplementary files

• Supplementary file 1. Spreadsheet containing all strains, oligonucleotides and plasmids used, and how they were made.

• Supplementary file 2. Plasmid map of pFa6a-mCherry-ase1-kanMX6, mentioned in *Supplementary file 1* and in the Key Resources Table.

• Transparent reporting form

## Data availability

Source code to reproduce and analyse the simulations is deposited in github at https://github.com/manulera/simulationsLeraRamirez2021 (copy archived at swh:1:rev:f095ebc2b520d40c448ac-761361c568f7cd5f308). Source code to annotate and analyse kymographs is deposited in github at https://github.com/manulera/KymoAnalyzer (copy archived at swh:1:rev:a2915939f6031fb11448968b13fd8eecb363b2fe). Source code to find spindles in microscopy images is deposited in github at https://github.com/manulera/ImageAnalysisFunctions/tree/master/detection_functions/spindle (copy archived at swh:1:rev:f095ebc2b520d40c448ac761361c568f7cd5f308). We have uploaded the source data for all figures as comma separated values files.

The following datasets were generated:

| Author(s) | Year | Dataset title | Dataset URL | Database and Identifier |
|---|---|---|---|---|
| Ramirez ML | 2021 | Microtubule rescue at midzone edges promotes overlap stability and prevents spindle collapse during anaphase B | https://github.com/manulera/simulationsLeraRamirez2021 | GitHub, simulationsLeraRamirez2021 |
| Ramirez ML | 2021 | Microtubule rescue at midzone edges promotes overlap stability and prevents spindle collapse during anaphase B | https://github.com/manulera/ImageAnalysisFunctions/tree/master/detection_functions/spindle | GitHub, detection_functions/spindle |

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

# Appendix 1

To study the effect of 'condition' (genetic background or drug treatment) and 'position' (before / outside / inside, see cartoons in *Figure 3*) on microtubule growth speed, the data was fitted to the following mixed-effects model (in MATLAB / R notation): `speed condition * position + (condition|experiment) + (position|cell:experiment)` In which we consider the fixed effects of 'position' and 'condition' and their interaction. We also include random effects associated with experiment and cell variability. `(condition|experiment)` represents that the effect of 'condition' may vary between experiments, and `(position|cell:experiment)` represents that the effect of 'position' may vary between cells (which are nested within an experiment). For each of the conditions tested, we present three tables:

1. The top table contains the summary of a two-way anova, in which D.F. num., D.F. denom are the degrees of freedom of the numerator and denominator estimated with Satterthwaite approximation.
2. The middle table presents the mean and 95% confidence interval for all combinations of 'position' and 'condition'.
3. The bottom table contains the summary of the mixed effect model, with the values of coefficients, their 95% confidence interval and statistical significance.

For more details on statistical analysis, see Methods.

**Appendix 1—table 1.** Summary statistics for klp9$^{OE}$ and cdc25-22.

| Category | D.F. num | D.F. denom | F-Statistic | p-value |
|---|---|---|---|---|
| position | 2.00 | 78.60 | 127.63 | 2.1x10–25 |
| condition | 2.00 | 3.26 | 0.41 | 0.69 |
| position:condition | 4.00 | 110.65 | 0.54 | 0.71 |

| Condition | Position | Mean 95 % C.I. |
|---|---|---|
| wt | before | 1.54 ± 0.34 µm/min |
| wt | outside | 1.32 ± 0.30 µm/min |
| wt | inside | 0.63 ± 0.35 µm/min |
| klp9$^{OE}$ | before | 1.66 ± 0.19 µm/min |
| klp9$^{OE}$ | outside | 1.53 ± 0.19 µm/min |
| klp9$^{OE}$ | inside | 0.82 ± 0.27 µm/min |
| cdc25-22 | before | 1.58 ± 0.12 µm/min |
| cdc25-22 | outside | 1.34 ± 0.24 µm/min |
| cdc25-22 | inside | 0.74 ± 0.10 µm/min |

| Coefficient | Estimate 95 % C.I. | T-statistic | p-value |
|---|---|---|---|
| Intercept (wt before) | 1.54 ± 0.34 µm/min | 15.94 | 1.3x10-3 |
| outside | –0.22 ± 0.14 µm/min | –3.11 | 2.2x10-3 |
| inside | –0.91 ± 0.12 µm/min | –15.86 | 5.0x10–21 |
| klp9$^{OE}$ | 0.12 ± 0.49 µm/min | 0.78 | 0.49 |
| cdc25-22 | 0.04 ± 0.39 µm/min | 0.36 | 0.75 |
| outside & klp9$^{OE}$ | 0.09 ± 0.21 µm/min | 0.82 | 0.41 |
| inside & klp9$^{OE}$ | –0.03 ± 0.19 µm/min | –0.30 | 0.76 |
| outside & cdc25-22 | 0.08 ± 0.21 µm/min | 0.72 | 0.47 |
| inside & cdc25-22 | 0.06 ± 0.17 µm/min | 0.73 | 0.47 |

**Appendix 1—table 2.** Summary statistics for ark1-as3.

| Category | D.F. num | D.F. denom | F-Statistic | p-value |
|---|---|---|---|---|
| position | 2.00 | 59.50 | 70.39 | 2.1x10–16 |
| condition | 2.00 | 5.40 | 0.095 | 0.91 |
| position:condition | 4.00 | 123.60 | 4.62 | 1.6x10-3 |

| Condition | Position | Mean 95 % C.I. |
|---|---|---|
| wt | before | 1.49 ± 0.13 µm/min |
| wt | outside | 1.31 ± 0.15 µm/min |
| wt | inside | 0.79 ± 0.13 µm/min |
| ark1-as3 normal | before | 1.49 ± 0.28 µm/min |
| ark1-as3 normal | outside | 1.23 ± 0.29 µm/min |
| ark1-as3 normal | inside | 0.84 ± 0.33 µm/min |
| ark1-as3 abnormal | before | 1.45 ± 0.28 µm/min |
| ark1-as3 abnormal | outside | 1.41 ± 0.31 µm/min |
| ark1-as3 abnormal | inside | 1.24 ± 0.28 µm/min |

| Coefficient | Estimate 95 % C.I. | T-statistic | p-value |
|---|---|---|---|
| Intercept (wt before) | 1.49 ± 0.13 µm/min | 27.53 | 1.8x10-8 |
| outside | –0.18 ± 0.13 µm/min | –2.70 | 9.3x10-3 |
| inside | –0.71 ± 0.12 µm/min | –11.85 | 5.0x10–18 |
| ark1-as3 normal | –0.00 ± 0.25 µm/min | –0.039 | 0.97 |
| ark1-as3 abnormal | –0.04 ± 0.27 µm/min | –0.40 | 0.71 |
| outside & ark1-as3 normal | –0.08 ± 0.25 µm/min | –0.63 | 0.53 |
| inside & ark1-as3 normal | 0.14 ± 0.19 µm/min | 1.47 | 0.15 |
| outside & ark1-as3 abnormal | 0.06 ± 0.22 µm/min | 0.56 | 0.58 |
| inside & ark1-as3 abnormal | 0.49 ± 0.23 µm/min | 4.14 | 7.2x10-5 |

**Appendix 1—table 3.** Summary statistics for cerulenin.

| Category | D.F. num | D.F. denom | F-Statistic | p-value |
|---|---|---|---|---|
| position | 2.00 | 27.70 | 82.08 | 2.3x10–12 |
| condition | 1.00 | 8.11 | 7.83 | 0.023 |
| position:condition | 2.00 | 41.22 | 2.00 | 0.15 |

| Condition | Position | Mean 95 % C.I. |
|---|---|---|
| ctrl | before | 1.50 ± 0.18 µm/min |
| ctrl | outside | 1.36 ± 0.18 µm/min |
| ctrl | inside | 0.60 ± 0.17 µm/min |
| cerulenin | before | 1.66 ± 0.10 µm/min |
| cerulenin | outside | 1.16 ± 0.57 µm/min |
| cerulenin | inside | 0.37 ± 0.44 µm/min |

| Coefficient | Estimate 95 % C.I. | T-statistic | p-value |
|---|---|---|---|
| Intercept (ctrl before) | 1.50 ± 0.18 µm/min | 27.02 | 1.4x10-4 |
| inside | –0.90 ± 0.14 µm/min | –12.79 | 2.5x10–13 |
| outside | –0.14 ± 0.14 µm/min | –2.03 | 0.046 |

*Continued on next page*

*Continued*

| Coefficient | Estimate 95 % C.I. | T-statistic | p-value |
|---|---|---|---|
| cerulenin | 0.16 ± 0.13 µm/min | 2.80 | 0.023 |
| inside & cerulenin | −0.38 ± 0.46 µm/min | −1.71 | 0.100 |
| outside & cerulenin | −0.36 ± 0.58 µm/min | −1.22 | 0.23 |

**Appendix 1—table 4.** Summary statistics for ase1Δ.

| Category | D.F. num | D.F. denom | F-Statistic | p-value |
|---|---|---|---|---|
| position | 2.00 | 46.89 | 27.26 | $1.4 \times 10^{-8}$ |
| condition | 1.00 | 26.80 | 15.26 | $5.7 \times 10^{-4}$ |
| position:condition | 2.00 | 56.53 | 4.81 | 0.012 |

| Condition | Position | Mean 95 % C.I. |
|---|---|---|
| wt | before | 1.49 ± 0.21 µm/min |
| wt | outside | 1.32 ± 0.21 µm/min |
| wt | inside | 0.71 ± 0.22 µm/min |
| ase1Δ | before | 1.79 ± 0.19 µm/min |
| ase1Δ | outside | 1.56 ± 0.19 µm/min |
| ase1Δ | inside | 1.34 ± 0.20 µm/min |

| Coefficient | Estimate 95 % C.I. | T-statistic | p-value |
|---|---|---|---|
| Intercept (wt before) | 1.49 ± 0.21 µm/min | 16.56 | $1.6 \times 10^{-7}$ |
| inside | −0.78 ± 0.22 µm/min | −7.28 | $1.1 \times 10^{-8}$ |
| outside | −0.17 ± 0.13 µm/min | −2.62 | 0.012 |
| ase1Δ | 0.30 ± 0.16 µm/min | 3.91 | $5.7 \times 10^{-4}$ |
| inside & ase1Δ | 0.33 ± 0.29 µm/min | 2.27 | 0.029 |
| outside & ase1Δ | −0.06 ± 0.19 µm/min | −0.67 | 0.51 |

**Appendix 1—table 5.** Summary statistics for les1Δ.

| Category | D.F. num | D.F. denom | F-Statistic | p-value |
|---|---|---|---|---|
| position | 2.00 | 42.54 | 88.84 | $6.5 \times 10^{-16}$ |
| condition | 1.00 | 9.32 | 1.65 | 0.23 |
| position:condition | 2.00 | 41.41 | 2.45 | 0.099 |

| Condition | Position | Mean 95 % C.I. |
|---|---|---|
| wt | before | 1.61 ± 0.10 µm/min |
| wt | outside | 1.51 ± 0.14 µm/min |
| wt | inside | 0.83 ± 0.09 µm/min |
| les1Δ | before | 1.72 ± 0.20 µm/min |
| les1Δ | outside | 1.63 ± 0.22 µm/min |
| les1Δ | inside | 1.13 ± 0.21 µm/min |

| Coefficient | Estimate 95 % C.I. | T-statistic | p-value |
|---|---|---|---|
| Intercept (wt before) | 1.61 ± 0.10 µm/min | 33.63 | $3.1 \times 10^{-19}$ |
| inside | −0.78 ± 0.12 µm/min | −13.10 | $1.2 \times 10^{-17}$ |
| outside | −0.10 ± 0.14 µm/min | −1.50 | 0.14 |
| les1Δ | 0.11 ± 0.20 µm/min | 1.28 | 0.23 |

*Continued on next page*

*Continued*

| Coefficient | Estimate 95 % C.I. | T-statistic | p-value |
|---|---|---|---|
| inside & les1Δ | 0.19 ± 0.18 μm/min | 2.14 | 0.038 |
| outside & les1Δ | 0.02 ± 0.21 μm/min | 0.15 | 0.88 |

**Appendix 1—table 6.** Summary statistics for nem1Δ.

| Category | D.F. num | D.F. denom | F-Statistic | p-value |
|---|---|---|---|---|
| position | 2.00 | 99.50 | 182.11 | 5.6x10–34 |
| condition | 1.00 | 4.57 | 0.36 | 0.58 |
| position:condition | 2.00 | 98.48 | 2.61 | 0.079 |

| Condition | Position | Mean 95 % C.I. |
|---|---|---|
| wt | before | 1.65 ± 0.22 μm/min |
| wt | outside | 1.45 ± 0.19 μm/min |
| wt | inside | 0.70 ± 0.20 μm/min |
| nem1Δ | before | 1.69 ± 0.11 μm/min |
| nem1Δ | outside | 1.50 ± 0.12 μm/min |
| nem1Δ | inside | 0.90 ± 0.12 μm/min |

| Coefficient | Estimate 95 % C.I. | T-statistic | p-value |
|---|---|---|---|
| Intercept (wt before) | 1.65 ± 0.22 μm/min | 26.84 | 4.1x10-4 |
| inside | −0.95 ± 0.10 μm/min | −18.63 | 4.9x10–29 |
| outside | −0.20 ± 0.10 μm/min | −3.90 | $1.2 \times 10^{-4}$ |
| nem1Δ | 0.04 ± 0.16 μm/min | 0.60 | 0.58 |
| inside & nem1Δ | 0.16 ± 0.15 μm/min | 2.09 | 0.040 |
| outside & nem1Δ | 0.01 ± 0.16 μm/min | 0.17 | 0.87 |

