## [Editor Report]

This study carefully quantifies microtubule dynamics during anaphase in the fission yeast *S. pombe*. The high quality data revealed two novel observations: that microtubule rescue occurs preferentially at the edge of the midzone and that microtubule growth speed decreases when the nuclear membrane wraps around the spindle midzone in late anaphase. This sheds additional light on the interplay between the nuclear membrane and the midspindle in closed mitosis, and this study will be of interest to cell biologists studying spindle dynamics and mitosis.

---

## [Decision Letter]

**Decision letter after peer review:**

Thank you for submitting your article "Microtubule rescue at midzone edges promotes overlap stability and prevents spindle collapse during anaphase B" for consideration by *eLife*. Your article has been reviewed by 3 peer reviewers, and the evaluation has been overseen by a Reviewing Editor and Anna Akhmanova as the Senior Editor. The reviewers have opted to remain anonymous.

The reviewers have discussed their reviews with one another, and the Reviewing Editor has drafted this letter to help you prepare a revised submission.

You will find the public reviews by all three reviewers below. The section here summarizes what we consider to be essential revisions for publication in *eLife*.

In addition, this section contains suggestions that are not essential for publication but that may strengthen the paper and which you may want to consider.

Essential revisions:

1) You conclude that Ase1 is required for the slowing of microtubule growth speed. However, this conclusion relies on a small number of data points from the few cells that form long spindles in ase1D cells (Figure 1F).

(i) Please add a statistical analysis to support your claim.

(ii) What is the relation between spindles and nuclear membrane bridges in ase1D cells? Which fraction of cells reaches a spindle length, where the midspindle will be encapsulated by the nuclear membrane? Can you show the absence of slowing specifically in such cells?

2) Please add statistical analyses for other crucial data, such as Figure 2G-L and similar graphs in the supplement.

3) Given that the results in this paper rely heavily on Alp7 tagged with 3xGFP, please provide information on the impact (or not) of the tag on microtubule dynamics.

4) It was confusing to several reviewers that the statement "transition from fast to slow microtubule growth occurs in the absence of known anaphase MAPs" (e.g. title of Figure 2) is followed by showing that Ase1, which is an anaphase MAP, is in fact required. This will be confusing to readers as well. Please change the phrasing (minimally the title of Figure 2 and the corresponding subheading).

Suggested (but not essential) revisions:

5) With respect to the effect of the nuclear membrane bridge on microtubule growth speed, it would be interesting to determine the microtubule polymerization dynamics in an imp1D strain.

6) Better presentation of the results (please strongly consider making these changes)

– The kymographs would be easier to interpret if they were centered on the geometric center of the spindle.

– In Figure 2, which graphs present normalized data should be more clearly indicated in the figure. Or, alternatively, the figure could either show only normalized data, and the supplementary figure only non-normalized data.

– In Figure 1 and 2, the 'distance between the plus end and the closest pole at time of rescue' is used to represent progression through anaphase. Measures such as 'spindle length' or 'time since onset of anaphase B' may be more intuitive to most readers.

– Figure 4I may be easier to understand if all wt, all ark1-as3 normal and all ark1-as3 abnormal were grouped together. (This will make it easier to understand the ark1-as3 abnormal cells, where the 'outside' and 'inside' classes are different from those in the other two situations.)

– Figure 5: This kymograph appears to elongate similar to wt cells and reach a long length. Is it indeed representative of ase1D cells?

– In Figure 5D, the rescue data could be represented in a similar way as Figure 1G to link the results back to these earlier experiments.

– Figure 6G: The caption states that spindle collapse occurred, but the graphic does not show it.

– Since only the microtubule ends are labeled, microtubules cannot be unambiguously assigned to a spindle pole. How microtubules are assigned to one of the two spindle poles, and which assumptions are made to do this, should be more clearly described. This is crucial since the information is essential to determine whether a feature in the kymographs represents a growing microtubule from the distal pole or a shrinking microtubule from the proximal pole. The data for ase1∆ mutants in Figure 5 amplifies this concern.

– Description of the implementation of the model could be improved in the methods part, and some essential features should additionally be mentioned in the text. For example, is Ase1 the only crosslinker in the system? What is the source of spindle elongation force in the model?

7) Additional analysis of existing data

– Figure 1 focuses on growth rate and rescue. The shrinking rate and catastrophe frequency could be additionally presented, either in the figure or in a table.

– The experiments in Figure 2 could be examined for rescue (see Public Review by reviewer #2).

8) Emphasis in different sections, additional discussion, and relation to previous work:

– The introduction would profit from including additional information on closed mitosis.

– The discussion could contain an additional part that discusses if the reduction in microtubule growth speed at anaphase B serves a specific function, or is merely a by-product of nuclear division.

– The discussion might also benefit from returning to the bigger picture on the coordination/complementary roles of spindle sliding and polymerization in different systems.

– The papers mentioned by reviewer #2 in point 3 of the public review and their implications could be discussed.

– The discussion on mechanisms by which the nuclear membrane bridge may alter MT growth speed could include some of the possibilities mentioned by reviewer #1 in the public review.

– Since the ase1-off strain shows a reduction in microtubule growth speed, whereas the ase1D strain does not, it may be worth discussing that it cannot be excluded that a complete deletion of cls1 (which was not examined, since lethal) may also show a different phenotype from the cls1-off strain.

---

## [Author Response]

Essential revisions:1) You conclude that Ase1 is required for the slowing of microtubule growth speed. However, this conclusion relies on a small number of data points from the few cells that form long spindles in ase1D cells (Figure 1F).(i) Please add a statistical analysis to support your claim.(ii) What is the relation between spindles and nuclear membrane bridges in ase1D cells? Which fraction of cells reaches a spindle length, where the midspindle will be encapsulated by the nuclear membrane? Can you show the absence of slowing specifically in such cells?

We have repeated our analysis in cells expressing a nuclear membrane marker in order to be able to categorise the events as in Figure 3D and performed a statistical analysis (see Figure 5F). We have restricted the analysis to the cells that undergo dumbbell transition (see section “The decrease in growth speed […] upon ase1 deletion”).

We have found that the effect of ase1 deletion, while consistent with our previous claim, is not as strong as we originally proposed: Microtubule growth decreases when microtubules enter the nuclear membrane bridge in ase1D cells, but to a lesser extent than in wild-type cells (~25% decrease in ase1D vs. ~50% decrease in wt). We have changed the text, figure 5, and section titles accordingly.

2) Please add statistical analyses for other crucial data, such as Figure 2G-L and similar graphs in the supplement.

This comment refers to the figure panels where we have represented microtubule growth speed as a function of distance to the closest pole. We acknowledge that the old Figure 5F could be insufficient to support our claims that ase1 deletion prevents the decrease in microtubule growth speed, especially considering that we used data both from cells that reached and did not reach the dumbbell transition. We have addressed this in the previous point, and used a membrane marker to categorise events and compare them.

For the other panels (Figure 2 and supplements, supplement of Figure 3, and supplement of Figure 5), we have added the statistical significance of difference between means in each bin (see Methods).

3) Given that the results in this paper rely heavily on Alp7 tagged with 3xGFP, please provide information on the impact (or not) of the tag on microtubule dynamics.

We have compared the measurements of microtubule growth speed, growth event duration and rescue distribution in cells expressing Alp7-3xGFP and GFP-Mal3 (EB1 analogue). Cells expressing GFP-Mal3 have similar microtubule dynamics during interphase when compared to cells expressing labelled α tubulin (Busch et al., 2004), and GFP-tagged EB proteins are commonly used for measuring microtubule dynamics in multiple systems. We have found that microtubule dynamics are similar in both strains. The results are summarised in Figure 1 – Supplement 2, and in the main text, in the section “Anaphase B microtubule dynamics are similar in cells expressing GFP-Mal3 and Alp7-3xGFP”.

In addition, following comments from Crowd Review, we have also used the GFP-Mal3 data to compare microtubule growth speed and growth event duration in interphase and anaphase in the same conditions. Before, we had mentioned that in our dataset (using Alp7-3xGFP) both growth speed and growth event duration during anaphase B were higher than what was reported in the literature using labelled tubulin. This was because Alp7-3xGFP does not strongly label interphase microtubule plus ends. In Figure 1 – Supplement 2F, G we show that when measuring them in the same conditions using GFP-Mal3, this difference is also observed.

4) It was confusing to several reviewers that the statement "transition from fast to slow microtubule growth occurs in the absence of known anaphase MAPs" (e.g. title of Figure 2) is followed by showing that Ase1, which is an anaphase MAP, is in fact required. This will be confusing to readers as well. Please change the phrasing (minimally the title of Figure 2 and the corresponding subheading).

We agree that this title could be misleading. We have switched the title and header from “known anaphase MAPs” to “important MAPs” to avoid confusion. To explain why we chose these candidates, we have added the sentence “we next examined various candidate MAPs associated with spindle microtubules that have been shown or proposed to regulate microtubule dynamics during anaphase B”

Suggested (but not essential) revisions:5) With respect to the effect of the nuclear membrane bridge on microtubule growth speed, it would be interesting to determine the microtubule polymerization dynamics in an imp1D strain.

It has been shown in Lucena et al., 2015 that Imp1 is involved in the disassembly of the nuclear membrane at the end of mitosis, which is required for spindle disassembly. Consequently, in imp1D cells the mitotic spindle elongates beyond the wild-type final length, until the membrane bridge is cleaved by the cytokinetic ring. Data from Dey et al., 2020 suggests that Imp1 is involved in the sequential removal of nuclear pore complexes from the membrane, which eventually leads to nuclear envelope breakdown, allowing cytoplasmic factors to disassemble the spindle.

Our original expectation was that in imp1D cells microtubules inside the nuclear membrane bridge would elongate with lower speed, as in the wild-type, and would remain in this slow growth state for longer, due to the delay in spindle disassembly. Surprisingly, we found that, although microtubule growth velocity decreased as in the wild-type, it eventually increased again at late anaphase (see Author response image 1, dashed line at B, and D, E), at spindle lengths where the spindle would have disassembled in wild-type cells (Author response image 1). This was particularly evident in the rare cases where imp1D spindles did not go out of focus when spindle poles reached the cell tips and late anaphase spindles could be observed for a long period of time (Author response image 1, right kymograph at B).

**Author response image 1. sa2fig1:** (A-B) Kymographs of anaphase B mitotic spindles in cells expressing Alp7-3xGFP, Sid4-GFP (top) and Cut11-mCherry (bottom) in wild-type (A) and imp1 deleted (B) cells. Dashed line indicates increase in microtubule growth speed at late anaphase. Arrowheads in (B) mark a cluster of Cut11-mCherry (left) or higher intensity of Cut11-mCherry at the center of the spindle. (C) Time-lapse images of imp1 deleted cells expressing Alp7-3xGFP, Sid4-GFP (top) and Cut11-mCherry (bottom). Time between images is 3 minutes, scalebar 3 μm. Empty arrowheads mark the appearance of the Post Anaphase Array (PAA) (D-E) Microtubule growth speed as a function of spindle length (D) and distance to closest pole (E) at rescue (or first point if rescue could not be exactly determined). Microtubule growth events of clear duration are shown as round dots; other events are shown as stars. Thick lines represent average of binned data, error bars represent 95% confidence interval of the mean. p-values represent statistical significance of the difference of means with respect to wild-type at each bin (see Methods). “n.s” (not significant) indicates p>0.05. Number of observations: 431 (33 cells) wt, 549 (36 cells) imp1*Δ* microtubule growth events from 4 independent experiments. We have uploaded the source data with the manuscript.

It should be noted that in some imp1D cells, we observed stronger signal of cut11-mCherry at the nuclear bridge than in wild-type (Author response image 1, arrows on right kymograph at B), consistent with the role of Imp1 in removing nuclear pore complexes from the membrane. In other imp1D cells, we observed clusters of Cut11-mCherry which were not present in wild-type cells (Author response image 1, arrow on left kymograph at B, and cut11-mCherry channel in C). Additionally, the Post Anaphase Array (PAA, a microtubule structure nucleated from the cytokinetic actin ring at the end of anaphase) was formed before spindle disassembly in imp1D cells and remained in presence of the spindle for several minutes (Author response image 1, empty arrowheads in C), unlike in wild-type cells, where the formation of the PAA roughly coincides with spindle disassembly.

Further experiments would be required to understand the imp1D phenotype. There could be multiple reasons why microtubule growth increases in imp1D cells at late anaphase. The different nuclear membrane composition (evidenced by higher signal or clusters of cut11-mCherry), or the formation of the PAA might perturb the interaction between spindle microtubules and the nuclear membrane specifically at late anaphase. PAA microtubules could sequester a required for this interaction, leading to an increase in microtubule growth. Alternatively, a partial fenestration of the nuclear membrane bridge produced by the cytokinetic ring might be sufficient to increase microtubule growth speed, without inducing the disassembly of the spindle. Finally, it could be produced by signalling triggered once the spindle reaches a certain length, as proposed recently for mitotic exit signalling in animal cells (Alfonso et al., 2019). The fact that the decrease in microtubule growth velocity is completely inhibited if dumbbell transition does not occur (ark1-as3 abnormal and cerulenin in Figure 4), and delayed in time in klp9D cells, which have a delayed dumbbell transition (Figure 2 Supp. 1B) strongly argues against a cell cycle regulation of microtubule growth during anaphase, as discussed extensively in the paper.

We believe these results are interesting, and deserve further exploration in the future. However, we would like to not include this data in the main text, as it does not change the conclusions we originally presented, and may diverge the attention from the main messages of the paper. Ultimately, our data clearly shows that Imp1 is not required for the decrease in microtubule growth associated with the formation of the nuclear membrane bridge (Author response image 1).

6) Better presentation of the results (please strongly consider making these changes)– The kymographs would be easier to interpret if they were centered on the geometric center of the spindle.

We have centered all the kymographs.

– In Figure 2, which graphs present normalized data should be more clearly indicated in the figure. Or, alternatively, the figure could either show only normalized data, and the supplementary figure only non-normalized data.

We have changed the representation. Instead of showing normalised data, we have added a scaled y axis for the non-wt condition on the right side of the plot (See Figure 2 H, J). This is equivalent to normalization, since both axis start at y=0, but this representation keeps the axis on the left equivalent to the ones in non-normalised plots.

– In Figure 1 and 2, the 'distance between the plus end and the closest pole at time of rescue' is used to represent progression through anaphase. Measures such as 'spindle length' or 'time since onset of anaphase B' may be more intuitive to most readers.

The distance between the plus end and the closest pole at time of rescue represents both anaphase progression and position within the spindle, as this distance increases for all microtubules as the spindle elongates, but is higher for microtubules that are rescued closer to the center for a given spindle length. Ultimately, we show that this magnitude is a proxy for whether a microtubule is rescued inside or outside the nuclear bridge, and that is why it separates the two populations of points better than spindle length / anaphase progression (Figure 3E).

We agree that the introduction to this measurement without the membrane context might have been insufficient and have improved the text on this, adding also a quantitative argument (value of R2) as to why this parameter is a better predictor. Interestingly, the R2 is very similar when using a fit to a step function of speed vs. rescue respect to pole or when using the before / inside / outside categorisation, further arguing for these methods being equivalent.

We believe that for the story, it is preferable to rule out candidates that have been previously proposed to govern microtubule growth speed during anaphase (Figure 2), and then bring up the novel concept of the membrane bridge being important (Figure 3).

– Figure 4I may be easier to understand if all wt, all ark1-as3 normal and all ark1-as3 abnormal were grouped together. (This will make it easier to understand the ark1-as3 abnormal cells, where the 'outside' and 'inside' classes are different from those in the other two situations.)

We have represented the data as suggested.

– Figure 5: This kymograph appears to elongate similar to wt cells and reach a long length. Is it indeed representative of ase1D cells?

This is indeed a representative kymograph of the ase1D condition, but not all spindles reach the normal final length, as shown in Figure 5D. For completeness, we have included an example where the spindle collapses early, Figure 5B right, and referred to it in the text when we mention that spindles collapse in ase1D.

– In Figure 5D, the rescue data could be represented in a similar way as Figure 1G to link the results back to these earlier experiments.

We have represented it as a histogram.

– Figure 6G: The caption states that spindle collapse occurred, but the graphic does not show it.

In our simulations, the forces in the system are not simulated (sliding occurs at a constant speed imposed in the simulation). Therefore, we define spindle collapse as the point when there is no overlap between antiparallel microtubules, at which point the simulation stops. To better show that the ends of the kymographs in 6G correspond to spindle collapses we have added a small graph representing the spindle at the last point of the kymographs in 6D and G. We have also clarified this better in the methods section.

This is likely related to the fact that the model was not sufficiently well presented in the main text (see point 6i in this document).

– Since only the microtubule ends are labeled, microtubules cannot be unambiguously assigned to a spindle pole. How microtubules are assigned to one of the two spindle poles, and which assumptions are made to do this, should be more clearly described. This is crucial since the information is essential to determine whether a feature in the kymographs represents a growing microtubule from the distal pole or a shrinking microtubule from the proximal pole. The data for ase1∆ mutants in Figure 5 amplifies this concern.

We have included this in the method section (see Image and data analysis > kymograph analysis).

– Description of the implementation of the model could be improved in the methods part, and some essential features should additionally be mentioned in the text. For example, is Ase1 the only crosslinker in the system? What is the source of spindle elongation force in the model?

We have improved the method description, and have made it more clear in the methods and main text that we do not specifically simulate crosslinkers nor forces, and that we instead assume a constant elongation speed of the spindle, and consider the midzone to be a region of constant length at the spindle center.

7) Additional analysis of existing data– Figure 1 focuses on growth rate and rescue. The shrinking rate and catastrophe frequency could be additionally presented, either in the figure or in a table.

We have added univariate distribution plots of this measurements to Figure 1-Supplement 1 and referred to them in the main text. We have also included a table with the values of microtubule dynamics (Table 1).

– The experiments in Figure 2 could be examined for rescue (see Public Review by reviewer #2).

We have added a supplementary figure representing the rescue distribution in the different conditions, and a reference to it in the main text (Figure 2 – Supplement 4).

8) Emphasis in different sections, additional discussion, and relation to previous work:– The introduction would profit from including additional information on closed mitosis.

We have added some additional introduction covering the roles of Les1, which is relevant for the results of the paper.

– The discussion could contain an additional part that discusses if the reduction in microtubule growth speed at anaphase B serves a specific function, or is merely a by-product of nuclear division.

We have added this to the discussion.

– The discussion might also benefit from returning to the bigger picture on the coordination/complementary roles of spindle sliding and polymerization in different systems.

We agree that the end was abrupt and we realised that it would be better to finish with the main message of the paper. Therefore, we have switched the order of the last two sections of the discussion, since the last one not only contains the main message, but also sets up better for returning to the bigger picture (see last paragraph of discussion).

– The papers mentioned by reviewer #2 in point 3 of the public review and their implications could be discussed.

Regarding the effect of PRC1 on spindle elongation during anaphase, and the different results from Tolic and Kapoor papers, we agree with the comment from the reviewer. Our original intent in "… depletion of PRC1/Ase1 does not perturb anaphase spindle elongation" was simply to say that PRC1 depletion does not lead to spindle collapse, rather than to say that it does not affect microtubule growth velocity. We have changed the text to reflect this more explicitly.

We thank the Reviewers for pointing us to Asthana et al., as the last published version also shows that microtubule growth decreases with anaphase progression in RPE-1 cells, which we have mentioned in the introduction.

We have mentioned the interaction of Bim1 and Ase1 described in Thomas et al., 2020 in the context of the distribution of rescues in mal3D cells (Figure 2 – Supp. 4). Regarding the similarity between our ase1D data and the data in this publication, the authors show that ase1 truncation ase1∆^693^ (which prevents binding of Bim1/EB1 to Ase1) leads to spindle collapse. However, the authors do not report spindle collapse for ase1^SHNN^, a mutant that specifically disrupts Bim1-Ase1 interactions, suggesting that something else in the C-term of Ase1 might be responsible for spindle collapse. The turnover of PRC1 in vitro is increased when its C-terminal is removed (Subramanian et al., 2010), so ase1∆^693^ may be unable to form a stable midzone due to high turnover.

– The discussion on mechanisms by which the nuclear membrane bridge may alter MT growth speed could include some of the possibilities mentioned by reviewer #1 in the public review.

We believe our data and previous observations suggest that fenestration does not mediate the change in microtubule growth that we observe.

Local disassembly of the nuclear membrane occurs at late anaphase: in Dey et al., 2020 NLS-GFP leakage from the nucleus in cells that fail to form a tight seal at the nuclear bridge occurs once the spindle is fully elongated. On the other hand, the decrease in microtubule growth speed that we observe occurs at mid-anaphase, coinciding with the dumbbell transition.

Additionally, our few observations of cerulenin treated cells where the dumbbell transition occurred after a substantial period of elongation arrest (Figure 4H) showed an abrupt change in microtubule growth speed, suggesting that the wrapping of the membrane alone was sufficient to induce the change in microtubule growth speed, unless fenestration occurred immediately.

Our imp1D data, (mentioned in this response letter) shows that in the absence of Imp1, which prevents the disassembly of the nuclear membrane, the decrease in microtubule growth speed occurs with similar dynamics, suggesting that the disassembly of nuclear membrane is not required for microtubule growth speed to decrease. The fact that the microtubule growth speed eventually increases in imp1D cells might indicate that a partial fenestration mediated by the cytokinetic ring restores the early anaphase microtubule growth speed, but more experiments would be required to confirm this.

Finally, the reviewer suggests that our observation that microtubule growth is not decreased inside the ectopic tubes formed at the spindle edges in ark1-as3 cells might indicate that fenestration is required for microtubule growth speed to decrease. The nuclear membrane in these tubes is likely very different from the nuclear membrane bridge. Importantly, these tubes do not have the spindle-nuclear membrane interactions mediated by Ase1 mentioned in Exposito-Serrano et al., 2020.

– Since the ase1-off strain shows a reduction in microtubule growth speed, whereas the ase1D strain does not, it may be worth discussing that it cannot be excluded that a complete deletion of cls1 (which was not examined, since lethal) may also show a different phenotype from the cls1-off strain.

We have addressed this in the main text.